# Transition Matching:
# Scalable and Flexible Generative Modeling

**Neta Shaul**[*,1,†]        **Uriel Singer**[*,2]        **Itai Gat**[2]        **Yaron Lipman**[2]

[1]Weizmann Institute of Science, [2]FAIR at Meta,
[†]Work done during internship at Meta FAIR, [*]Joint first author

## Abstract

Diffusion and flow matching models have significantly advanced media generation, yet their design space is well-explored, somewhat limiting further improvements. Concurrently, autoregressive (AR) models, particularly those generating continuous tokens, have emerged as a promising direction for unifying text and media generation. This paper introduces Transition Matching (TM), a novel discrete-time, continuous-state generative paradigm that unifies and advances both diffusion/flow models and continuous AR generation. TM decomposes complex generation tasks into simpler Markov transitions, allowing for expressive non-deterministic probability transition kernels and arbitrary non-continuous supervision processes, thereby unlocking new flexible design avenues. We explore these choices through three TM variants: (i) Difference Transition Matching (DTM), which generalizes flow matching to discrete-time by directly learning transition probabilities, yielding state-of-the-art image quality and text adherence as well as improved sampling efficiency. (ii) Autoregressive Transition Matching (ARTM) and (iii) Full History Transition Matching (FHTM) are partially and fully causal models, respectively, that generalize continuous AR methods. They achieve continuous causal AR generation quality comparable to non-causal approaches and potentially enable seamless integration with existing AR text generation techniques. Notably, FHTM is the first fully causal model to match or surpass the performance of flow-based methods on text-to-image task in continuous domains. We demonstrate these contributions through a rigorous large-scale comparison of TM variants and relevant baselines, maintaining a fixed architecture, training data, and hyperparameters.

## 1   Introduction

Recent progress in diffusion models and flow matching has significantly advanced media generation (images, video, audio), achieving state-of-the-art results [31, 24, 34, 4]. However, the design space of these methods has been extensively investigated [43, 20, 30, 40, 7], potentially limiting further significant improvements with current modeling approaches. An alternative direction focuses on autoregressive (AR) models to unify text and media generation. Earlier approaches generated media as sequences of discrete tokens either in raster order [37, 54, 6]; or in random order [3]. Further advancement was shown by switching to continuous token generation [25, 47], while also improving performance at scale [10].

This paper introduces Transition Matching (TM), a general discrete-time continuous-state generation paradigm that unifies diffusion/flow models and continuous AR generation. TM aims to advance both paradigms and create new state-of-the-art generative models. Similar to diffusion/flow models, TM breaks down complex generation tasks into a series of simpler Markov transitions. However, unlike diffusion/flow, TM allows for expressive non-deterministic probability transition kernels and arbitrary non-continuous supervision processes, offering new and flexible design choices.

39th Conference on Neural Information Processing Systems (NeurIPS 2025).

| FM | MAR | FHTM (Ours) | DTM (Ours) |
|---|---|---|---|

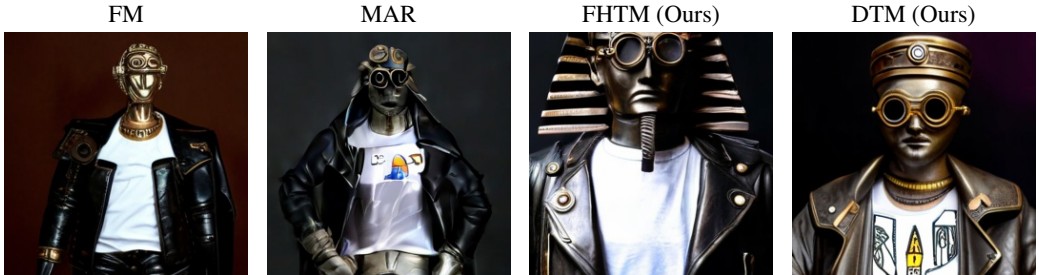

*"A portrait of a metal statue of a pharaoh wearing steampunk glasses and a leather jacket over a white t-shirt that has a drawing of a space shuttle on it."*

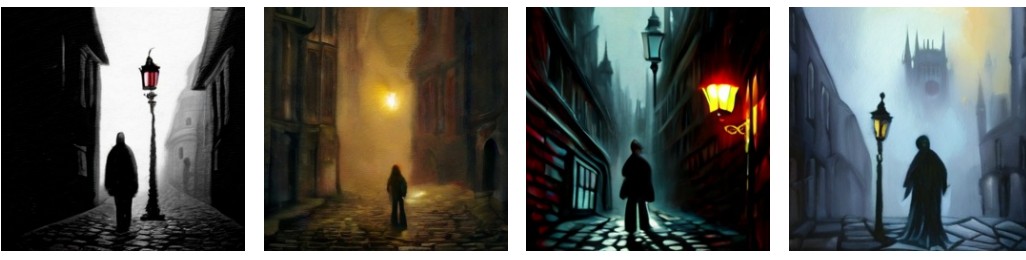

*"A solitary figure shrouded in mists peers up from the cobble stone street at the imposing and dark gothic buildings surrounding it. an old-fashioned lamp shines nearby. oil painting."*

Figure 1: Transition Matching methods (FHTM and DTM) compared to baselines (FM and MAR) under a fixed architecture, dataset and training hyper-parameters.

We explore these design choices and present three TM variants:

(i) Difference Transition Matching (DTM): A generalization of flow matching to discrete time, DTM directly learns the transition probabilities of consecutive states in the linear (Cond-OT) process instead of just its expectation. This straightforward approach yields a state-of-the-art generation model with improved image quality and text adherence, as well as significantly faster sampling.

(ii) Autoregressive Transition Matching (ARTM) and (iii) Full History Transition Matching (FHTM): These partially and fully causal models (respectively) generalize continuous AR models by incorporating a multi-step generation process guided by discontinuous supervising processes. ARTM and FHTM achieve continuous causal AR generation quality comparable to non-causal methods. Importantly, their causal nature allows for seamless integration with existing AR text generation methods. FHTM is the first fully causal model to match or surpass the performance of flow-based methods in continuous domains.

In summary, our contributions are:

1. Formulating transition matching: simplified and generalized discrete-time generative models based on matching transition kernels.

2. Identifying and exploring key design choices, specifically the supervision process, kernel parameterization, and modeling paradigm.

3. Introducing DTM, which improves upon state-of-the-art flow matching in image quality, prompt alignment, and sampling speed.

4. Introducing ARTM and FHTM: partially and fully causal AR models (resp.) that match non-AR generation quality and state-of-the-art prompt alignment.

5. Presenting a fair, large-scale comparison of the different TM variants and relevant baselines using a fixed architecture, data, and training hyper-parameters.

## 2 Transition Matching

We start by describing the framework of Transition Matching (TM), which can be seen as a simplified and general discrete time formulation for diffusion/flow models. Then, we focus on several, unexplored TM design choices and instantiations that goes beyond diffusion/flow models. In particular: we consider more powerful transition kernels and/or discontinuous noise-to-data processes. In the experiments section we show these choices lead to state-of-the-art image generation methods.

### 2.1 General framework

**Notation** We use capital letters $X, Y, Z, A, B$ to denote random variables (RVs) and lower-case letter $x, y, z, a, b$ to denote their particular states. One exception is time $t$ where we abuse notation a bit and use it to denote both particular times and a RV. All our variables and states reside in euclidean spaces $x \in \mathbb{R}^d$. The probability density function (PDF) of a random variable $Y$ is denoted $p_Y(x)$. For RVs $X_t$ (and only for them) we use the simpler PDF notation $p_t(x_t)$. We use the standard notations for joints $p_{X,Y}(x, y)$ and conditional densities $p_{X|Y}(x|y)$ densities. We denote $[T] = \{0, 1, \ldots, T\}$.

**Problem definition** Given a training set of i.i.d. samples from an unknown *target distribution* $p_T$, and some easy to sample *source distribution* $p_0$. Our goal is to learn a Markov Process, defined by a *probability transition kernel* $p_{t+1|t}^\theta(x_{t+1}|x_t)$, where $t \in [T-1]$ taking us from $X_0 \sim p_0$ to $X_T \sim p_T$. That is, we define a series of random variables $(X_t)_{t \in [T]}$ such that $X_0 \sim p_0$ and

$$X_{t+1} \sim p_{t+1|t}^\theta(\cdot|X_t) \text{ for all } t \in [T-1] \text{ then } X_T \sim p_T. \tag{1}$$

**Supervising process** Training such a Markov process is done with the help of a *supervising process*, which is a stochastic process $(X_0, X_1, \ldots, X_T)$ defined given data samples $X_T$ using a *conditional process* $q_{0,\ldots,T-1|T}$, i.e.,

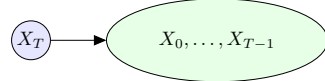

Figure 2: Supervising process.

$$q_{0,\ldots,T}(x_0, \ldots, x_T) = q_{0,\ldots,T-1|T}(x_0, \ldots, x_{T-1}|x_T)p_T(x_T), \tag{2}$$

and $q_{0,\ldots,T}$ denotes the joint probability of the supervising process $(X_t)_{t \in T}$. The only constraint on the conditional process is that its marginal at time $t = 0$ is the easy to sample distribution $p_0$, i.e.,

$$q_0 = p_0. \tag{3}$$

Note that this definition is very general and allows, for example, arbitrary non-continuous processes, and indeed we utilize such a process below. Transition matching engages with the supervising process $(X_t)_{t \in T}$ by sampling pairs of consecutive states $(X_t, X_{t+1}) \sim q_{t,t+1}$, $t \in [T-1]$.

**Loss** The model $p_{t+1|t}^\theta$ is trained to transition between consecutive states $X_t \to X_{t+1}$ in the sense of equation 1 by regressing $q_{t+1|t}$ defined from the supervising process $q$. This motivates the loss utilizing a distance/divergence $D$ between distributions

$$\mathcal{L}(\theta) = \mathbb{E}_{t,X_t} D\big(q_{t+1|t}(\cdot|X_t), p_{t+1|t}^\theta(\cdot|X_t)\big), \tag{4}$$

where $t$ is sampled uniformly from $[T-1]$. However, this loss requires evaluating $q_{t+1|t}$ which is usually hard to compute. Therefore, to make the training tractable we require that the distance $D$ has an *empirical form*, i.e., can be expressed as an expectation of an empirical one-sample loss $\hat{D}$ over target samples. We define the loss

$$\mathcal{L}(\theta) = \mathbb{E}_{t,X_t} \overbrace{\mathbb{E}_{X_{t+1}} \hat{D}(X_{t+1}, p_{t+1|t}^\theta(\cdot|X_t))}^{D \text{ in empirical form}} = \mathbb{E}_{t,X_t,X_{t+1}} \hat{D}\big(X_{t+1}, p_{t+1|t}^\theta(\cdot|X_t)\big), \tag{5}$$

where $(X_t, X_{t+1})$ are sampled from the joint $q_{t,t+1}$ with the help of equation 2, namely, first sample data $X_T \sim p_T$ and then $(X_t, X_{t+1}) \sim q_{t,t+1|T}(\cdot|X_T)$. Notably, equation 5 can be used to learn arbitrary transition kernels, in contrast to e.g., Gaussian kernels used in discrete time diffusion models or deterministic kernels used in flow matching. The particular choice of the cost $D$ depends on the modeling paradigm chosen for the transition kernel $p_{t+1|t}^\theta$, and discussed later.

| **Algorithm 1** Transition Matching Training | **Algorithm 2** Transition Matching Sampling |
|---|---|
| **Require:** $p_T$ ▷ Data | **Require:** $p_0$ ▷ Source distribution |
| **Require:** $q_{t,Y\|T}$ ▷ Process | **Require:** $p_{Y\|t}^\theta$ ▷ Trained model |
| **Require:** $T$ ▷ Number of TM steps | **Require:** $q_{t+1\|t,Y}$ ▷ Parametrization |
| 1: **while** not converged **do** | **Require:** $T$ ▷ Number of TM steps |
| 2:  Sample $t \sim \mathcal{U}([T-1])$, $X_T \sim p_T$ | 1: Sample $X_0 \sim p_0$ |
| 3:  Sample $(X_t, Y) \sim q_{t,Y\|T}(\cdot\|X_T)$ | 2: **for** $t = 0$ to $T - 1$ **do** |
| 4:  $\mathcal{L}(\theta) \leftarrow \hat{D}(Y, p_{Y\|t}^\theta(\cdot\|X_t))$ | 3:  Sample $Y \sim p_{Y\|t}^\theta(\cdot\|X_t)$ |
| 5:  $\theta \leftarrow \theta - \gamma \nabla_\theta \mathcal{L}$ ▷ Optimization step | 4:  Sample $X_{t+1} \sim q_{t+1\|t,Y}(\cdot\|X_t, Y)$ |
| 6: **end while** | 5: **end for** |
| 7: **return** $\theta$ | 6: **return** $X_T$ |

**Kernel parameterizations** The first and natural option to parameterize $p_{t+1|t}^\theta$ is to regress $q_{t+1|t}$ directly as is done in equation 5. This turns out to be a good modeling choice in certain cases. However, in some cases one can use other parameterizations that turn out to be beneficial, as is also done for flow and diffusion models. To do that in the general case, we use the *law of total probability* applied to the conditional probabilities $q_{t+1|t}$ with some latent RV $Y$:

$$q_{t+1|t}(x_{t+1}|x_t) = \int q_{t+1|t,Y}(x_{t+1}|x_t, y)q_{Y|t}(y|x_t)\mathrm{d}y, \tag{6}$$

where $q_{Y|t}$ is the *posterior distribution* of $Y$ given $X_t = x_t$ and $q_{t+1|t,Y}$ is easy to sample (often a deterministic function of $X_t$ and $Y$). Then the posterior of $Y$ is set as the new target of the learning process instead of the transition kernel. That is, instead of the loss in equation 5 we consider

$$\mathcal{L}(\theta) = \mathbb{E}_{t,X_t,Y}\hat{D}\big(Y, p_{Y|t}^\theta(\cdot|X_t)\big). \tag{7}$$

Similarly, during training, sampling from the joint $(X_t, Y) \sim q_{t,Y}$, is accomplished by first sampling data $X_T$ and then $(X_t, Y) \sim q_{t,Y|T}(\cdot|X_t)$. Once the posterior $p_{Y|t}^\theta$ is trained, sampling from the transition $p_{t+1|t}^\theta$ during inference is done with the help of equation 6. To summarize, in cases we want to use non-trivial kernel parameterization, i.e., $Y \neq X_{t+1}$, we sample from $q_{t+1|t,Y}$ (in sampling) and $q_{t,Y|T}$ (in training). See Algorithms 1 and 2 the training and sampling pseudocodes.

**Kernel modeling** Once a desirable $Y$ is identified, the remaining part is to choose a generative model for the kernel $p_{Y|t}^\theta$. Importantly, one of the key advantages in TM comes from choosing expressive kernels that result in more elaborate transition kernels than used previously. A kernel modeling is set by a choice of a *probability model* for $p_{Y|t}^\theta$ and a *loss* to learn it. We denote the probability model choice by $B|A$, where $A$ denotes the *condition* and $B$ the *target*. For example, $Y|X_t$ will denote a model that predicts a sample of $Y$ given a sample of $X_t$. We will also use more elaborate probability models, such as autoregressive models. To this end, consider the state $Y$ reshaped into individual *tokens* $Y = (Y^1, \ldots, Y^n)$, and then $Y^i \big| \big(Y^{<i}, X_t\big)$ means that our model samples the token $Y^i$ given previous tokens of $Y$, $Y^{<i} = (Y^1, \ldots, Y^{i-1})$, and $X_t$.

All our models are learned with flow matching (FM) loss. For completeness, we provide the key components of flow matching formulated generically to learn to sample from $B|A$. Individual states of $A$ and $B$ are denoted $a$ and $b$, respectively. Flow matching models $p_{B|A}^\theta$ via a velocity field $u_s^\theta(b|a)$ that is used to sample from $p_{B|A}^\theta(\cdot|a)$ by solving the Ordinary Differential Equation (ODE)

$$\frac{\mathrm{d}B_s}{\mathrm{d}s} = u_s^\theta(B_s|a) \tag{8}$$

initializing with a sample $B_0 \sim \mathcal{N}(0, I)$ (the standard normal distribution) and solving until $s = 1$. In turn, $B_1$ is the desired sample, i.e., $B_1 \sim p_{B|A}^\theta(\cdot|a)$. The loss $D$, used to train FM, has an empirical form and minimizes the difference between $q_{B|A}$ and $p_{B|A}^\theta$,

$$\hat{D}\big(B, p_{B|A}^\theta(\cdot|a)\big) = \mathbb{E}_{s,B_0} \big\| u_s^\theta(B_s|a) - (B - B_0) \big\|^2, \tag{9}$$

where $s$ is sampled uniformly in $[0, 1]$, $B_0 \sim \mathcal{N}(0, I)$, $B \sim q_{B|A}(\cdot|a)$, and $B_s = (1 - s)B_0 + sB$.

We summarize the key design choices in Transition Matching:

| TM design: | Supervising process | Parametrization | Modeling |
|---|---|---|---|
| | $q$ | $Y$ | $B\|A$ |

## 2.2 Transition Matching made practical

The key contribution of this paper is identifying previously unexplored design choices in the TM framework that results in effective generative models. We focus on two TM variants: Difference Transition Matching (DTM), and Autoregressive Transition Matching (ARTM/FHTM).

**Difference Transition Matching**    Our first instance of TM makes the following choices:

| DTM: | Supervising process | Parametrization | Modeling |
|---|---|---|---|
| | $X_t$ linear | $Y = X_T - X_0$ | $B = Y \| A = (t, X_t)$ |

As the supervising process $q$ we use the standard *linear process* (a.k.a., Conditional Optimal Transport), defined by

$$X_t = \left(1 - \frac{t}{T}\right) X_0 + \frac{t}{T} X_T, \qquad t \in [T], \tag{10}$$

where $X_0 \sim p_0 = \mathcal{N}(0, I)$ and $X_T \sim p_T$. This is the same process used in [27, 28]. For the kernel parameterization $Y$ we will use the *difference latent* (see Figure 3, left),

$$Y = X_T - X_0. \tag{11}$$

During training, sampling $q_{t,Y|T}(\cdot|X_T)$ (i.e., given $X_T$) is done by sampling $X_0$, and using 10 and 11 to compute $X_t, Y$. Using the definition in 10 and rearranging gives

$$X_{t+1} = X_t + \frac{1}{T}Y, \tag{12}$$

and this equation can be used to sample from $q_{t+1|t,Y}(\cdot|X_t, Y)$ during inference. See Figure 4 for an illustration of a sampled path from this supervising process. We learn to sample from the posterior $p_{Y|t}^\theta \approx q_{Y|t}$ using flow matching with $A = (t, X_t)$ and $B = Y$. This means we learn a velocity field $u_s^\theta(y|t, x_t)$ and train it with Algorithm 1 and the CFM loss in equation 9.

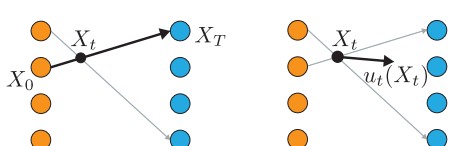

Figure 3: Difference prediction given $X_t$ (left) and flow matching velocity $u_t(X_t)$ (right).

Note that in this case one can also learn a continuous time $t \in [0, T]$ which allows more flexible sampling.

The last remaining component is choosing the architecture for $u_s^\theta$. Let $x = (x^1, \ldots, x^n)$ be a reshaped state to $n$ tokens. For example, each $x^i$ can represent a patch in an image $x$. Next, note that in each transition step we need to sample $Y \sim p_{Y|t}^\theta(\cdot|X_t)$ by approximating the solution of the ODE in equation 8. Therefore, to keep the sampling process efficient, we follow [25] and use a small *head* $g^\theta$ that generates all tokens in a batch and is fed with latents from a large *backbone* $f^\theta$. Our velocity model is defined as

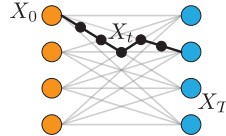

Figure 4: DTM path sampled with eq. 12.

$$u_s^\theta(y|t, x_t) = \left[g_{s,t}^\theta(y^1, h_t^1), \ldots, g_{s,t}^\theta(y^n, h_t^n)\right], \tag{13}$$

where $h_t^i$ is the $i$-th output token of the backbone, i.e., $[h_t^1, h_t^2, \ldots, h_t^n] = f_t^\theta(x_t)$. See Figure 5 (DTM) for an illustration of this architecture. One limitation of this architecture worth mentioning is that in each transition step, each token $y^i$ is generated independently, which limits the power of this kernel. We discuss this in the experiments section but nevertheless demonstrate that DTM with this architecture still leads to state-of-the-art image generation model.

**Connection to flow matching**    Although flow matching [27, 28, 1] is a *deterministic* process while DTM samples from a stochastic transition kernel in each step, a connection between the two is revealed by noting that the *expectation* of a DTM step coincides with Flow Matching Euler step, i.e.,

$$\mathbb{E}[Y|X_t = x] = \mathbb{E}[X_T - X_0|X_t = x] = u_t(x), \tag{14}$$

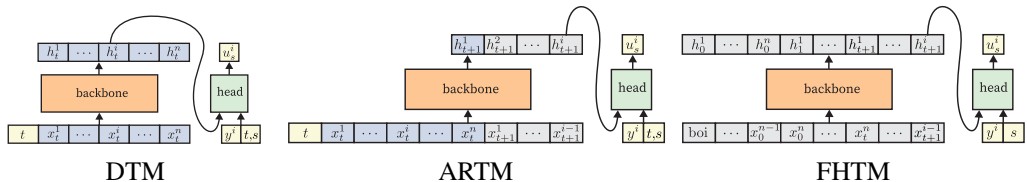

Figure 5: Architectures of the methods suggested in the paper. Backbone (orange) is the main network (transformer); head (green) is a small network (2% backbone parameters); blue tokens use full attention, gray tokens are causal; $u_s^i$ is the output velocity.

which is exactly the marginal velocity in flow matching, see Figure 3. In fact, as $T \to \infty$ (or equivalently, steps are getting smaller), DTM is becoming more and more deterministic, converging to FM with Euler step, providing a novel and unexpected elementary proof (i.e., without the continuity equation) for FM marginal velocity. In Appendix C we prove

**Theorem 1.** *(informal) As the number of steps increases, $T \to \infty$, DTM converges to Euler step FM,*

$$X_{t+k} \approx x_t + \frac{k}{T}\mathbb{E}\left[X_T - X_0 | X_t = x_t\right],$$

*as $k/T \to 0$, where $X_\ell$, $\forall \ell > t$ is defined by Algorithm 2 with a optimally trained $p_{Y|t}^\theta$.*

We attribute the empirical success of DTM over flow matching to its more elaborate kernel.

**Autoregressive Transition Matching**    Our second instance of TM is geared towards incorporating state-of-the-art media generation in autoregressive models, and utilizes the following choices:

| **ARTM:** | **Supervising process** | **Parametrization** | **Modeling** |
|---|---|---|---|
| | $X_t$ independent linear | $Y = X_{t+1}$ | $B = Y^i \,|\, A = (t, X_t, X_{t+1}^{<i})$ |

In this case we use a novel supervising process we call *independent linear process*, defined by

$$X_t = \left(1 - \frac{t}{T}\right)X_{0,t} + \frac{t}{T}X_T, \ t \in [T], \tag{15}$$

where $X_{0,t} \sim \mathcal{N}(0, I)$, $t \in [T]$ are all i.i.d. samples. Sampling $q_{t,t+1|T}(\cdot|X_T)$ is done by sampling $X_{0,t}$ and $X_{0,t+1}$ and using 15. Although the independent linear process has the same marginals $q_t$ as the linear process in equation 10, it enjoys better regularity of the conditional $q_{t+1|t}(\cdot|x_t)$, see Figure 6 for an illustration, and as demonstrated later in experiments is key for building state-of-the-art Autoregressive image generation models.

For the transition kernel we use an Autoregressive (AR) model with the choice of $Y = X_{t+1}$. As before, we let a state written as series of tokens $x = (x^1, \ldots, x^n)$ and write the target kernel $q_{t+1|t}$ using the probability chain rule (as usual in AR modeling),

$$q_{t+1|t}(X_{t+1}|X_t) = \prod_{i=1}^{n} q_{t+1|t}^i(X_{t+1}^i|X_t, X_{t+1}^{<i}),$$

where $X_{t+1}^{<1}$ is the empty state. We will learn to

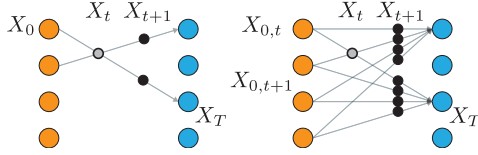

Figure 6: Linear process (left) and independent linear process (right) showing possible $X_{t+1}$ given a sample $X_t$. The independent process has much wider support for $X_{t+1}$ given $X_t$.

sample from $q_{t+1|t}^i$ using FM with $A = (t, X_t, X_{t+1}^{<i})$ and $B = X_{t+1}^i$. That is, we learn a velocity field $u_s^\theta(y^i|t, x_t, x_{t+1}^{<i})$ trained with the CFM loss in equation 9. This method builds upon the initial idea [25] that uses such AR modeling to map in a single transition step from $X_0$ to $X_T$ using diffusion, and in that sense ARTM is a generalization of that method. Lastly the architecture for $u_s^\theta$ is based on a similar construction to DTM with a few, rather minor changes. Using the same notation for the head $g^\theta$ and backbone $f^\theta$ models we define

$$u_s^\theta(y^i|t, x_t, x_{t+1}^{<i}) = g_{s,t}^\theta(y^i, h_{t+1}^i), \tag{16}$$

with $h_{t+1}^i = f_t^\theta(x_t, x_{t+1}^{<i})$. Figure 5 (ARTM) shows an illustration of this architecture.

**Full-History ARTM** We consider a variant of ARTM that allows full "teacher-forcing" training and consequently provides a good candidate to be incorporated into multimodal AR model.

| **FHTM:** | **Supervising process** | **Parametrization** | **Modeling** |
|---|---|---|---|
| | $X_{\leq t}$ independent linear | $Y = X_{t+1}$ | $B = Y^i \,|\, A = (X_0, \ldots, X_t, X_{t+1}^{<i})$ |

The idea is to use the full history of states, namely considering the kernel

$$q_{t+1|0,\ldots,t}(X_{t+1}|X_0,\ldots,X_t) = \prod_{i=1}^{n} q_{t+1|0,\ldots,t}^i(X_{t+1}^i|X_0,\ldots,X_t,X_{t+1}^{<i}), \tag{17}$$

and train an FM sampler from $q_{t+1|0,\ldots,t}^i$ with the choices $A = (X_0, \ldots, X_t, X_{t+1}^{<i})$ (no need to add time $t$ due to the full state sequence) and $B = X_{t+1}^i$. The architecture of the velocity $u_s$ is defined by

$$u_s^\theta(y^i|x_0,\ldots,x_t,x_{t+1}^{<i}) = g_s^\theta(y^i, h_{t+1}^i), \tag{18}$$

with $h_{t+1}^i = f^\theta(x_0,\ldots,x_t,x_{t+1}^{<i})$ and we take $f$ to be fully causal. See Figure 5 (FHTM).

## 3 Related work

**Diffusion and flows** We draw the connection to previous works from the perspective of transition matching. Diffusion models [41, 16, 42, 21] can be seen as an instance of TM by choosing $D$ in the loss (5) to be the KL divergence, derived in diffusion literature as the variational lower bound [22]. The popular $\epsilon$-prediction [16] in transition matching formulation is achieved by the design choices

| **$\epsilon$-prediction:** | **Supervising process** | **Parametrization** | **Modeling** |
|---|---|---|---|
| | $X_t = \sigma_t X_0 + \alpha_t X_T$ | $Y = X_0$ | $Y|X_t \sim \mathcal{N}\left(Y \,|\, \epsilon_t^\theta(X_t), w_t^2 I\right)$ |

where $(\sigma_t, \alpha_t)$ is the scheduler, and non-zero $w_t$ reproduces the sampling algorithm in [16], while taking the limit $w_t \to 0$ yields the sampling of [42]. Similarly, $x$-prediction [21] is achieved by the parametrization $Y = X_T$. In contrast to these work, our TM instantiations use more expressive kernel modeling. Relation to flow matching[27, 28, 1] is discussed in Section 2.2. Generator matching [17] generalizes diffusion and flow models to general continuous time Markov process modeled with arbitrary generators, while we focus on discrete time Markov processes. Another line of works adopted supervision processes that transition between different resolutions; [19] used flow matching with a particular coupling between different resolution as the kernel modeling; [56] implemented a similar scheme but allowed the FM to be dependent on the previous states (frames) in an AR manner. Denoising Diffusion GANs [50] uses $x$-prediction parametrization $Y = X_T$ and utilize a GAN [12] model as the transition kernel. In a concurrent work, [57] proposes a similar parameterization to DTM ($Y = X_T - X_0$) however uses a backbone-only architecture consequently making transition sampling computationally very expensive and sub-par in generation quality compared to the backbone-head architecture.

**Autoregressive image generation** Early progress in text-to-image generation was achieved using autoregressive models over discrete latent spaces [37, 8, 54], with recent advances [46, 44, 14] claiming to surpass flow-based approaches. A complementary line of work explores autoregressive modeling directly in continuous space [25, 47], demonstrating some advantages over discrete methods. In [10] this direction is scaled further, achieving SOTA results. In our experiments we compare these models in controlled setting and show that our autoregressive transition matching variants improves upon these models and achieves SOTA text-to-image performance with a fully causal architecture. Lastly, DART-AR [13] uses a supervising process similar to the independent linear process 15 with an autoregressive backbone however utilizing a Gaussian transition kernel per patch in contrast to an FM head used in our case.

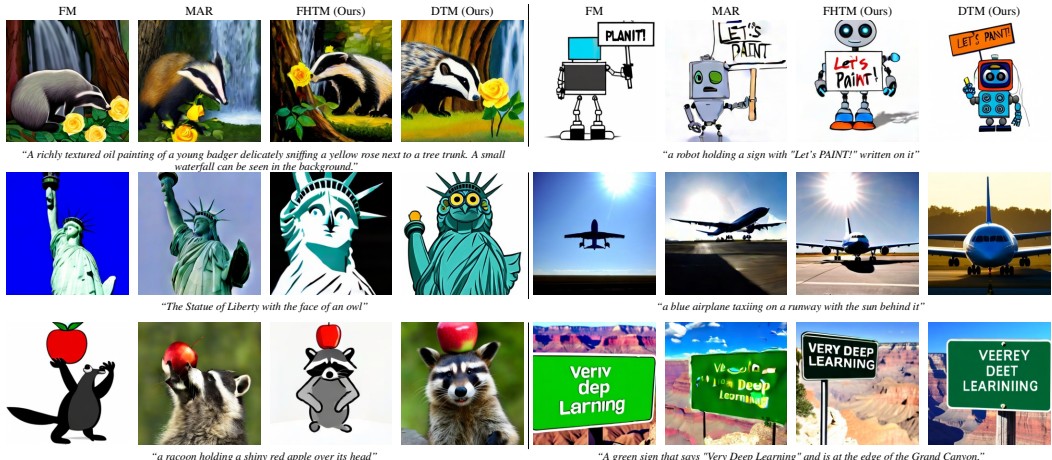

Figure 7: Samples comparison of our DTM, FHTM vs. FM, and MAR; Images were generated on similar DiT models trained for 1M iterations.

## 4 Experiments

We evaluate the performance of our Transition Matching (TM) variants—Difference TM (DTM), with $T = 32$ TM steps, Autoregressive TM (ARTM-2,3) with $T = 2, 3$ (resp.), and Full History TM (FHTM-2,3) with $T = 2, 3$ (resp.) — on the text-to-image generation task. In Appendix B we provide training and sampling pseudocodes of the three variants in Algorithms 3-8, and python code for training in Figures 25,26, and 27. Our baselines include flow matching (FM) [9], continuous-token autoregressive (AR) and masked AR (MAR) [25], and discrete-token AR [54] and MAR [3]. For continuous-token MAR we include two baselines: the original truncated Gaussian scheduler version [25], and the cosine scheduler used by Fluid (MAR-Fluid) [10].

**Datasets and metrics** Training dataset is a collection of 350M licensed Shutterstock image-caption pairs. Images are of $256 \times 256 \times 3$ resolution and captions span 1–128 tokens embedded with the CLIP tokenizer [35]. Consistent with prior work [38], for continuous state space, the images are embedded using the SDXL-VAE [33] into a $32 \times 32 \times 4$ latent space, and subsequently all model training are done within this latent space. For discrete state space, images are tokenized with Chameleon-VQVAE [2]. Evaluation datasets are PartiPrompts [54] and MS-COCO [26] text/image benchmarks. And the reported metrics are: CLIPScore [15], that emphasize prompt alignment; Aesthetics [39] and DeQA Score [53] that focus on image quality; PickScore [23], ImageReward [51], and UnifiedReward [49] which are human preference-based and consider both image quality and text adherence. Lastly, we report results on the GenEval [11] and T2I-CompBench [18] benchmarks.

**Architecture and optimization** All experiments are performed with the same 1.7B parameters DiT backbone ($f^\theta$) [32], excluding a single case in which we compare to a standard LLM architecture [48, 29]. Methods that require a small flow head ($g^\theta$), replace the final linear layer with a 40M parameters MLP [25]. Text conditioning is embedded through a Flan-UL2 encoder [45] and injected via cross attention layers, or as prefix in the single case of the LLM architecture. Finally, the models are trained for 500K iterations with a 2048 batch size. Precise details are in Appendix A.1. We aim to facilitate a fair and useful comparison between methods in large scale by fixing the training data, using the same size architectures with identical backbone (excluding the LLM architecture that use standard transformer backbone), and same optimization hyper-parameters. To this end, we restrict our comparison to baselines which we re-implemented.

### 4.1 Main results: Text-to-image generation

Our main evaluation results are reported in Tables 1 and 8 (in Appendix) on the DiT architecture. We find that DTM outperforms all baselines, and yields the best results across all metrics except the CLIPScore, where on the PartiPrompts benchmark it is a runner-up to MAR and our ARTM-3 and FHTM-3. On the MS-COCO benchmark, the discrete-state space models achieve the highest CLIPScore but lag behind on all other metrics, as well as on the GenEval benchmark. DTM

shows a considerable gain in text adherence over the baseline FM and sets a new SOTA on the text-to-image task. Next, our AR kernels with 3 TM steps: ARTM-3 and FHTM-3, demonstrate a significant improvement compared to the AR baseline, see comparison of samples in Figure 15 in the Appendix. When compared to MAR, ARTM-3 and FHTM-3 have comparable CLIPScore, but improve considerably on all other image quality metrics, where this is also noticeable qualitatively in Figure 7 and Figures 11-14 in the Appendix. GenEval and T2I-Compbench results are reported in Tables 2 and 9 (resp.) showing that overall DTM is leading with FHTM-3/ARTM-3 and MAR closely follows. To our knowledge, FHTM is the first fully causal model to match FM performance on text-to-image task in continuous domain, with improved text alignment.

Table 1: Evaluation of TM vs. baselines on PartiPrompts. $^\dagger$ Inference with activation caching. NFE$^*$ counts only backbone model evaluation ($f^\theta$). LLM and DiT have comparable number of parameters.

|  | Attention | Kernel | Arch | NFE$^*$ | CLIPScore ↑ | PickScore ↑ | ImageReward ↑ | UnifiedReward ↑ | Aesthetic ↑ | DeQA Score ↑ |
|---|---|---|---|---|---|---|---|---|---|---|
| Baseline | Full | MAR-discrete | DiT | 256 | 26.8 | 20.7 | 0.14 | 4.31 | 5.15 | 2.48 |
|  |  | MAR | DiT | 256 | 27.0 | 20.7 | 0.33 | 4.26 | 4.95 | 2.36 |
|  |  | MAR-Fluid | DiT | 256 | 26.0 | 20.5 | 0.07 | 3.82 | 4.74 | 2.36 |
|  |  | FM | DiT | 256 | 26.0 | 21.0 | 0.23 | 4.78 | 5.29 | 2.55 |
| TM |  | **DTM** | DiT | 32 | 26.8 | 21.2 | 0.53 | 5.12 | 5.42 | 2.65 |
| Baseline | Causal | AR-discrete$^\dagger$ | DiT | 256 | 26.7 | 20.4 | $-0.01$ | 3.74 | 4.81 | 2.38 |
|  |  | AR$^\dagger$ | DiT | 256 | 24.9 | 20.1 | $-0.43$ | 3.41 | 4.50 | 2.27 |
| TM |  | **ARTM$-$2**$^\dagger$ | DiT | $2 \times 256$ | 26.8 | 20.8 | 0.29 | 4.49 | 5.03 | 2.37 |
|  |  | **FHTM$-$2**$^\dagger$ | DiT | $2 \times 256$ | 26.8 | 20.8 | 0.30 | 4.59 | 5.13 | 2.44 |
|  |  | **ARTM$-$3**$^\dagger$ | DiT | $3 \times 256$ | 27.0 | 20.9 | 0.38 | 4.77 | 5.21 | 2.53 |
|  |  | **FHTM$-$3**$^\dagger$ | DiT | $3 \times 256$ | 27.0 | 20.9 | 0.31 | 4.77 | 5.15 | 2.44 |
|  |  | **FHTM$-$3**$^\dagger$ | LLM | $3 \times 256$ | 27.0 | 21.0 | 0.43 | 5.02 | 5.30 | 2.54 |

Table 2: Evaluation of TM versus baselines on GenEval; same settings as Table 1.

|  | Attention | Kernel | Arch | NFE$^*$ | Overall ↑ | Single-object ↑ | Two-objects ↑ | Counting ↑ | Colors ↑ | Position ↑ | Color Attribute ↑ |
|---|---|---|---|---|---|---|---|---|---|---|---|
| Baseline | Full | MAR-discrete | DiT | 256 | 0.44 | 0.86 | 0.43 | 0.37 | 0.66 | 0.13 | 0.29 |
|  |  | MAR | DiT | 256 | 0.52 | 0.98 | 0.56 | 0.43 | 0.73 | 0.11 | 0.38 |
|  |  | MAR-Fluid | DiT | 256 | 0.44 | 0.90 | 0.33 | 0.37 | 0.76 | 0.12 | 0.28 |
|  |  | FM | DiT | 256 | 0.47 | 0.91 | 0.52 | 0.27 | 0.71 | 0.12 | 0.34 |
| TM |  | **DTM** | DiT | 32 | 0.54 | 0.93 | 0.58 | 0.35 | 0.79 | 0.20 | 0.46 |
| Baseline | Causal | AR-discrete$^\dagger$ | DiT | 256 | 0.41 | 0.96 | 0.40 | 0.33 | 0.60 | 0.07 | 0.19 |
|  |  | AR$^\dagger$ | DiT | 256 | 0.34 | 0.86 | 0.26 | 0.15 | 0.63 | 0.06 | 0.15 |
| TM |  | **ARTM$-$2**$^\dagger$ | DiT | $2 \times 256$ | 0.49 | 0.95 | 0.51 | 0.39 | 0.79 | 0.11 | 0.27 |
|  |  | **FHTM$-$2**$^\dagger$ | DiT | $2 \times 256$ | 0.48 | 0.96 | 0.48 | 0.25 | 0.78 | 0.09 | 0.37 |
|  |  | **ARTM$-$3**$^\dagger$ | DiT | $3 \times 256$ | 0.51 | 0.95 | 0.54 | 0.41 | 0.79 | 0.16 | 0.28 |
|  |  | **FHTM$-$3**$^\dagger$ | DiT | $3 \times 256$ | 0.52 | 0.98 | 0.54 | 0.44 | 0.74 | 0.16 | 0.34 |
|  |  | **FHTM$-$3**$^\dagger$ | LLM | $3 \times 256$ | 0.49 | 0.94 | 0.55 | 0.37 | 0.69 | 0.17 | 0.29 |

**Image generation with causal model** Beyond improving prompt alignment and image quality in text-to-image task, a central goal of recent research [58, 55] is to develop multimodal models also capable of reasoning about images. This direction aligns naturally with our approach, as the fully causal FHTM variant enables seamless integration with large-language models (LLM) standard architecture, training, and inference algorithms. As a first step toward this goal, we demonstrate in Table 1 and 8 that FHTM, implemented with an LLM architecture replacing 2D with 1D positional encoding and input the text condition only at the first layer, can match

Table 3: Global ranking of TM and baselines on the benchmarks PartiPrompts, MS-COCO, GenEval, andT2I-CompBench; same settings as Table 1.

|  | Attention | Kernel | Arch | NFE$^*$ | Global rank ↓ |
|---|---|---|---|---|---|
| Baseline | Full | MAR-discrete | DiT | 256 | 200 |
|  |  | MAR | DiT | 256 | 127 |
|  |  | MAR-Fluid | DiT | 256 | 220 |
|  |  | FM | DiT | 256 | 179 |
| TM |  | **DTM** | DiT | 32 | 58 |
| Baseline | Causal | AR-discrete$^\dagger$ | DiT | 256 | 245 |
|  |  | AR$^\dagger$ | DiT | 256 | 321 |
| TM |  | **ARTM$-$2**$^\dagger$ | DiT | $2 \times 256$ | 184 |
|  |  | **FHTM$-$2**$^\dagger$ | DiT | $2 \times 256$ | 185 |
|  |  | **ARTM$-$3**$^\dagger$ | DiT | $3 \times 256$ | 130 |
|  |  | **FHTM$-$3**$^\dagger$ | DiT | $3 \times 256$ | 130 |
|  |  | **FHTM$-$3**$^\dagger$ | LLM | $3 \times 256$ | 99 |

and even surpass the performance of approximately the same size DiT architecture. Furthermore, it matches or improve upon all baselines across all metrics. Further implementation details are in Appendix A.1.

**Global ranking** As part of our main effort to empirically validate the Transition Matching framework, we trained 6 variants: DTM, ARTM-2/3, FHTM-2/3, and FHTM-3 (LLM) and 6 baselines: FM, MAR, MAR-Fluid, MAR-discrete, AR, AR-discrete. All models were evaluated on four benchmarks: PartiPrompts, GenEval, MS-COCO, and T2I-CompBench (Tables 1–9). In total, we considered 12 models and 27 metrics per model. To derive a single measure of overall performance, we assigned each model a rank from 1 (best) to 12 (worst) per metric and summed them across all benchmarks. Table 3 reports the global rank, where DTM substantially outperforms all other TM variants and baselines, followed by FHTM, ARTM, and MAR.

## 4.2 Evaluations

**Sampling efficiency** One important benefit in the DTM variant is its sampling efficiency compared to flow matching. In Table 10 we report CLIPScore and PickScore for DTM and FM for different numbers of backbone and head steps while in Table 11 we log the corresponding forward times. Notably, the number of backbone forwards in DTM sampling can be reduced con-

Table 4: FM and DTM sampling times.

| Kernel | time (sec) | CLIPScore | PickScore |
|--------|------------|-----------|-----------|
| FM | 10.8 | 26.0 | 21.0 |
| DTM | 1.6 | 26.8 | 21.1 |

siderably without sacrificing generation quality. Table 4 presents the superior sampling efficiency of DTM over FM: DTM achieves state-of-the-art results with only 16 backbone forwards, leading to an almost 7-fold speedup compared to FM, which requires 128 backbone forwards for optimal quality in this case. In contrast to DTM, ARTM/FHTM do not offer any speed-up, in fact they require backbone forwards equal to the number of transition steps times the number of image tokens, as specified in Tables 1,2,8; Figure 8 reports CLIPScore and PickScore for different number of head forwards which demonstrates that this number can be reduced up to 4 with some limited reduction in performance for ARTM/FHTM sampling.

**Dependent vs. independent linear process** To highlight the impact of the supervising process, we compare the linear process (10), where $X_0$ is sampled once for all $t \in [T]$, with the independent linear process (15), where $X_{0,t}$ is sampled for each $t \in [T]$ independently, on our autoregressive kernels: ARTM-3 and FHTM-3. The models are trained for 100K iterations and CLIPScore and PickScore are evaluated every 10K iterations. As shown in Figure 9, the independent linear process is far superior to the linear process on these kernels, see further discussion in Appendix A.4.

**DTM Kernel expressiveness** The DTM kernel (see equation 13) generates each token $y^i$ of dimension $2 \times 2 \times 4$, corresponding to an image patch, independently in each transition step. This architecture choice is done mainly for performance reasons to allow efficient transitions. In Figure 10 we compare performance using a higher dimension $y^i$, corresponding to a $2 \times 8 \times 4$ patches. As can be seen in these graphs, performance improves for this larger patch kernel for low number of transition steps (1-4 steps) and surprisingly stays almost constant for very low number of head step, up to even a single step. The fact that performance does not improve for larger number of transition steps can be partially explained with Theorem 1 that shows that larger number of steps result in a simpler transition kernel (which in the limit coincides with flow matching).

## 5 Conclusions

We introduce Transition Matching (TM), a novel generative paradigm that unifies and generalizes diffusion, flow and continuous autoregressive models. We investigate three instances of TM: DTM, which surpasses state-of-the-art flow matching in image quality and text alignment; and the causal ARTM and fully causal FHTM that achieve generation quality comparable to non-causal methods. The improved performance of ARTM/FHTM comes at the price of a higher sampling cost, i.e., NFE counts are proportional to the number of transition steps, see e.g., in Table 1. DTM, in contrast, requires less backbone forwards and leads to significant speed-up over flow matching sampling, see e.g., Table 4. Future research directions include improving the training and/or sampling via different time schedulers and distillation, as well as incorporating FHTM in a multimodal system. Our work does not introduce additional societal risks beyond those related to existing image generative models.

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

# A Experiments

## A.1 Implementation details

**DiT architecture**  The DiT architecture [32] uses 24 blocks of a self-attention layer followed by cross attention layer with the text embedding [36], with a 2048 hidden dimension, 16 attention heads, and utilize a 3D positional embedding [5]. Embedded image [33] size is $32 \times 32 \times 4$ and input to the DiT trough a patchify layer with patch size of $2 \times 2 \times 4$. The total number of parameters is 1.7B. Since ARTM gets as input both $X_t$ and $X_{t+1}$, which results in a longer sequence length, for a fair comparison across TM variations and baselines, for all other models we pad the input sequence to double its length.

**LLM architecture**  The LLM architecture [29] is similar to the DiT with the following differences: (i) time injection is removed, (ii) cross attention layer is removed and text embedding is input as a prefix (iii) it uses a simple 1D instead of 3D positional embedding. To compensate for reduction in number of parameters, we increase the number of self-attention layers to 34, reaching 1.7B total number of parameters (comparable to the DiT). FHTM-DiT gets $X_0$ as input but does not take a loss on it, while for FHTM-LLM we remove the $X_0$ all together and instead use a single *boi* (begin of image) token to save sequence length.

**Flow head architecture**  Following [25] we use an MLP with 6 layers and a hidden dimension of 1024. to convert from the backbone hidden dimension (2048) to the MLP hidden dimension (1024) we use a simple linear layer. Finally, we replace the time input with AdaLN[32] time injection.

**Optimization**  The models are trained for 500K iterations, with a 2048 total batch size, $1 * e^{-4}$ constant learning rate and 2K iterations warmup.

**Classifier free guidance**  To support classifier free guidance (CFG), during training, with probability of $0.15$, we drop the text prompt and replace it with empty prompt. Following [25], during sampling, we apply CFG to the velocity of the flow head ($g^\theta$) with a guidance scale of $6.5$.

## A.2 Main results: Text-to-image generation

**Additional Kernels and Baselines**  Similar to the extension of the AR kernel to ARTM, We extend the MAR-Fluid kernel to 2 and 3 transition steps, resulting with the **MARTM$-$2** and **MARTM$-$3** kernels. Furthermore, we investigate the performance of the *Restart* sampling algorithm [52] on the FM kernel, were noise is added during the sampling process. We follow the authors' suggestion and perform 1 restart from $t = 0.6$ to $t = 0.4$, 3 restarts from $t = 0.8$ to $t = 0.6$, and an additional 3 restarts from $t = 1$ to $t = 0.8$. The sampling is performed on a base of 1000 steps, resulting with a total of 2400 NFE. As an additional baseline, we sample the FM kernel with 2400 NFE. Results can be found in Tables 5,6,7.

Table 5: Evaluation of MARTM and the *Restart* sampling algorithm baselines on PartiPrompts.

| | Attention | Kernel | Arch | NFE* | CLIPScore ↑ | PickScore ↑ | ImageReward ↑ | UnifiedReward ↑ | Aesthetic ↑ | DeQAScore ↑ |
|---|---|---|---|---|---|---|---|---|---|---|
| | | MAR-Fluid | DiT | 256 | 26.0 | 20.5 | 0.07 | 3.82 | 4.74 | 2.36 |
| | | **MARTM$-$2** | DiT | 256 | 26.7 | 20.9 | 0.36 | 4.69 | 5.13 | 2.42 |
| | | **MARTM$-$3** | DiT | 256 | 26.4 | 20.9 | 0.25 | 4.48 | 5.11 | 2.49 |
| Baseline | Full | | | | | | | | | |
| | | FM | DiT | 256 | 26.0 | 21.0 | 0.23 | 4.78 | 5.29 | 2.55 |
| | | FM | DiT | 2400 | 26.0 | 21.1 | 0.24 | 4.81 | 5.29 | 2.55 |
| | | FM-*Restart* | DiT | 2400 | 26.1 | 21.1 | 0.34 | 4.83 | 5.31 | 2.53 |

**Flow head NFE**  We ablate the number of NFE required by the flow head ($g^\theta$) to reach best performance for each model. As shown in Figure 8, we observe the models reach saturation with relatively low NFE, and decide to report results on Tables 1, 8 and 2 with 64 NFE for the flow head.

**TM steps vs Flow head NFE for DTM**  We test the performance of the DTM variant as function of TM steps and Flow head NFE. As shown in Table 10, our DTM model achieve reach saturation about

Table 6: Evaluation of MARTM and the *Restart* sampling algorithm baselines on GenEval.

| Attention | Kernel | Arch | NFE* | Overall ↑ | Single-object ↑ | Two-objects ↑ | Counting ↑ | Colors ↑ | Position ↑ | Color Attribute ↑ |
|---|---|---|---|---|---|---|---|---|---|---|
| Baseline | Full | | | | | | | | | |
| | | MAR-Fluid | DiT | 256 | 0.44 | 0.90 | 0.33 | 0.37 | 0.76 | 0.12 | 0.28 |
| | | **MARTM−2** | DiT | 256 | 0.51 | 0.94 | 0.55 | 0.35 | 0.77 | 0.21 | 0.32 |
| | | **MARTM−3** | DiT | 256 | 0.52 | 0.91 | 0.58 | 0.41 | 0.77 | 0.14 | 0.38 |
| | | FM | DiT | 256 | 0.47 | 0.91 | 0.52 | 0.27 | 0.71 | 0.12 | 0.34 |
| | | FM | DiT | 2400 | 0.47 | 0.91 | 0.51 | 0.25 | 0.72 | 0.14 | 0.36 |
| | | FM-*Restart* | DiT | 2400 | 0.49 | 0.89 | 0.59 | 0.29 | 0.73 | 0.13 | 0.38 |

Table 7: Evaluation of MARTM and the *Restart* sampling algorithm baselines on MS-COCO.

| Attention | Kernel | Arch | NFE* | CLIPScore ↑ | PickScore ↑ | ImageReward ↑ | UnifiedReward ↑ | Aesthetic ↑ | DeQAScore ↑ |
|---|---|---|---|---|---|---|---|---|---|
| Baseline | Full | | | | | | | | |
| | | MAR-Fulid | DiT | 256 | 25.5 | 20.5 | −0.11 | 3.94 | 4.86 | 2.38 |
| | | **MARTM−2** | DiT | 256 | 25.9 | 21.0 | 0.17 | 4.93 | 5.33 | 2.41 |
| | | **MARTM−3** | DiT | 256 | 25.7 | 20.9 | 0.04 | 4.67 | 5.21 | 2.45 |
| | | FM | DiT | 256 | 25.8 | 21.1 | 0.09 | 5.00 | 5.45 | 2.47 |
| | | FM | DiT | 2400 | 25.8 | 21.1 | 0.09 | 5.00 | 5.45 | 2.47 |
| | | FM-*Restart* | DiT | 2400 | 25.8 | 21.1 | 0.15 | 5.11 | 5.48 | 2.44 |

16 TM steps and 4 Flow head steps, according to CLIPScore and PickScore. Generation time for a single image on a single H100 GPU is provided in Table 11.

Table 8: Evaluation of TM versus baselines on MS-COCO. [†] Inference is done with activation caching. NFE* counts only backbone model evaluation ($f^\theta$). LLM and DiT have comparable number of parameters.

| Attention | Kernel | Arch | NFE* | CLIPScore ↑ | PickScore ↑ | ImageReward ↑ | UnifiedReward ↑ | Aesthetic ↑ | DeQAScore ↑ |
|---|---|---|---|---|---|---|---|---|---|
| Baseline | Full | MAR-discrete | DiT | 256 | 26.6 | 20.6 | 0.01 | 4.14 | 5.27 | 2.41 |
| | | MAR | DiT | 256 | 26.1 | 20.7 | 0.17 | 4.62 | 5.06 | 2.34 |
| | | MAR-Fulid | DiT | 256 | 25.5 | 20.5 | −0.11 | 3.94 | 4.86 | 2.38 |
| | | FM | DiT | 256 | 25.8 | 21.1 | 0.09 | 5.00 | 5.45 | 2.47 |
| TM | | **DTM** | DiT | 32 | 26.2 | 21.2 | 0.22 | 5.38 | 5.55 | 2.58 |
| Baseline | | AR-discrete[†] | DiT | 256 | 26.7 | 20.3 | −0.06 | 3.83 | 4.93 | 2.34 |
| | | AR[†] | DiT | 256 | 24.8 | 20.1 | −0.48 | 3.60 | 4.76 | 2.34 |
| TM | Causal | **ARTM−2[†]** | DiT | 2 × 256 | 25.9 | 20.8 | 0.07 | 4.70 | 5.19 | 2.41 |
| | | **FHTM−2[†]** | DiT | 2 × 256 | 25.9 | 20.8 | 0.07 | 4.78 | 5.27 | 2.45 |
| | | **ARTM−3[†]** | DiT | 3 × 256 | 26.1 | 20.9 | 0.11 | 4.99 | 5.35 | 2.46 |
| | | **FHTM−3[†]** | DiT | 3 × 256 | 26.1 | 21.0 | 0.15 | 5.23 | 5.38 | 2.41 |
| | | **FHTM−3[†]** | LLM | 3 × 256 | 26.1 | 21.1 | 0.24 | 5.51 | 5.53 | 2.51 |

Table 9: Evaluation of TM versus baselines on T2I-CompBench; same settings as Table 8.

| | Attention | Kernel | Arch | NFE* | Color ↑ | Shape ↑ | Texture ↑ | 2D-Spatial ↑ | 3D-Spatial ↑ | Numeracy ↑ | Non-Spatial ↑ | Complex ↑ |
|---|---|---|---|---|---|---|---|---|---|---|---|---|
| Baseline | Full | MAR-discrete | DiT | 256 | 0.6666 | 0.4535 | 0.5316 | 0.1474 | 0.2693 | 0.4538 | 0.3090 | 0.3096 |
| | | MAR | DiT | 256 | 0.7378 | 0.5174 | 0.6588 | 0.1638 | 0.3002 | 0.4962 | 0.3082 | 0.3392 |
| | | MAR-Fluid | DiT | 256 | 0.6997 | 0.4768 | 0.6149 | 0.1454 | 0.2938 | 0.4681 | 0.3037 | 0.3289 |
| | | FM | DiT | 256 | 0.6855 | 0.4511 | 0.5615 | 0.1372 | 0.2706 | 0.4526 | 0.3026 | 0.3138 |
| TM | | **DTM** | DiT | 32 | 0.7316 | 0.4865 | 0.6597 | 0.1839 | 0.3113 | 0.5043 | 0.3075 | 0.3382 |
| Baseline | Causal | AR-discrete† | DiT | 256 | 0.6068 | 0.4757 | 0.5958 | 0.1095 | 0.2535 | 0.4423 | 0.3098 | 0.3097 |
| | | AR† | DiT | 256 | 0.5062 | 0.3669 | 0.5061 | 0.1041 | 0.2441 | 0.4210 | 0.2989 | 0.2983 |
| TM | | **ARTM−2**† | DiT | 2×256 | 0.6520 | 0.4430 | 0.5870 | 0.1475 | 0.2748 | 0.4800 | 0.3074 | 0.3267 |
| | | **FHTM−2**† | DiT | 2×256 | 0.6318 | 0.4318 | 0.5730 | 0.1403 | 0.2818 | 0.4830 | 0.3058 | 0.3229 |
| | | **ARTM−3**† | DiT | 3×256 | 0.6555 | 0.4738 | 0.5842 | 0.1459 | 0.2832 | 0.4855 | 0.3062 | 0.3227 |
| | | **FHTM−3**† | DiT | 3×256 | 0.6604 | 0.4640 | 0.5839 | 0.1394 | 0.2755 | 0.4810 | 0.3066 | 0.3223 |
| | | **FHTM−3**† | LLM | 3×256 | 0.6166 | 0.4618 | 0.5945 | 0.1688 | 0.3081 | 0.5010 | 0.3079 | 0.3310 |

## A.3 Sampling efficiency

Table 10: Performance of FM (c-d) and DTM (a-b) for different combinations of Head NFE and TM steps, computed on a subset of the PartiPrompts dataset (1024 out of 1632). Color intensity increases with higher performance.

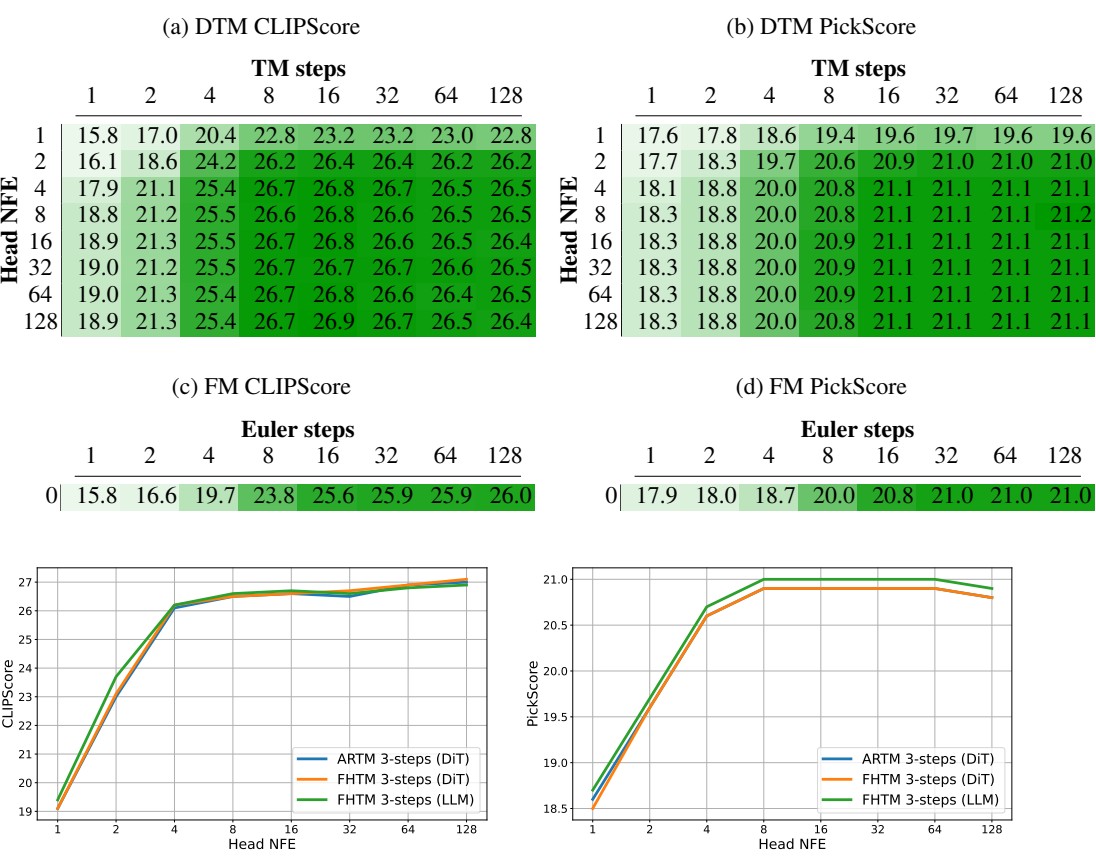

Figure 8: Comparison of flow head NFE vs. CLIPScore (left), and PickScore (right) computed on the PartiPrompts dataset.

Table 11: DTM inference time (in seconds) for different combinations of Head NFE and TM steps on a single H100 GPU. Color intensity increases with runtime. Note that 0 head steps refers to FM.

|  | TM steps (84 ms/step) | | | | | | | |
|---|---|---|---|---|---|---|---|---|
| Head NFE (3.5 ms/step) | 1 | 2 | 4 | 8 | 16 | 32 | 64 | 128 |
| 0 | 0.1 | 0.2 | 0.3 | 0.7 | 1.3 | 2.7 | 5.4 | 10.8 |
| 1 | 0.1 | 0.2 | 0.4 | 0.7 | 1.4 | 2.8 | 5.6 | 11.2 |
| 2 | 0.1 | 0.2 | 0.4 | 0.7 | 1.5 | 2.9 | 5.8 | 11.6 |
| 4 | 0.1 | 0.2 | 0.4 | 0.8 | 1.6 | 3.1 | 6.3 | 12.5 |
| 8 | 0.1 | 0.2 | 0.4 | 0.9 | 1.8 | 3.6 | 7.2 | 14.3 |
| 16 | 0.1 | 0.3 | 0.6 | 1.1 | 2.2 | 4.5 | 9.0 | 17.9 |
| 32 | 0.2 | 0.4 | 0.8 | 1.6 | 3.1 | 6.3 | 12.5 | 25.1 |
| 64 | 0.3 | 0.6 | 1.2 | 2.5 | 4.9 | 9.9 | 19.7 | 39.4 |
| 128 | 0.5 | 1.1 | 2.1 | 4.3 | 8.5 | 17.0 | 34.0 | 68.1 |

## A.4 Dependent vs. independent linear process

Further analysis of the generated images reveals that the AR kernels are unable to learn the linear process, resulting in low quality image generation. We hypothesize that the AR kernels exploit the linear relationship between $X_t$ and $X_{t+1}$ during training, which leads the model to learn a degenerate function and causes it to fail in inference.

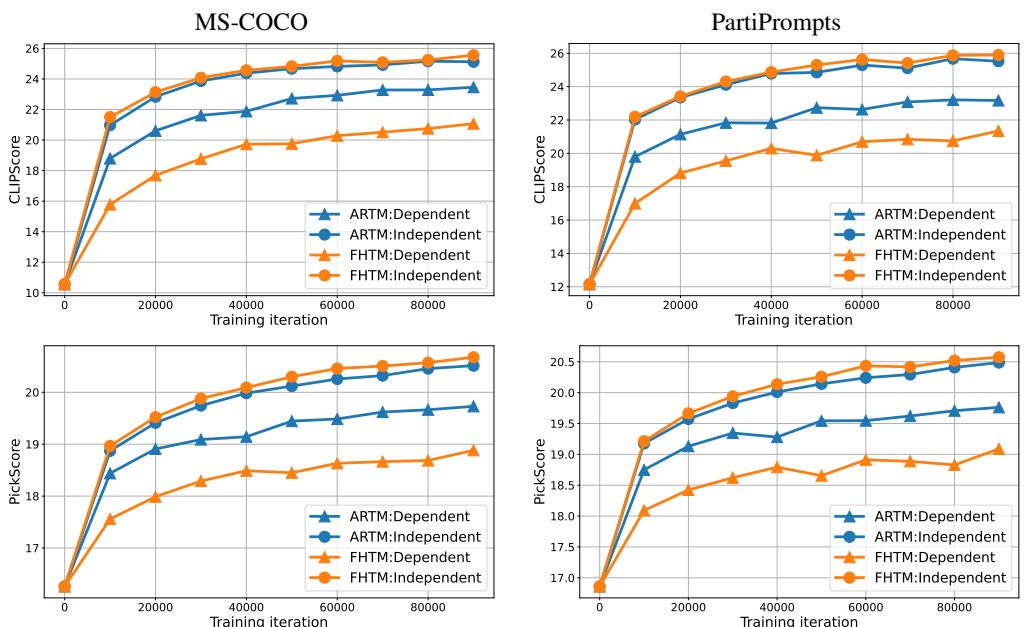

Figure 9: Dependent linear process (10) vs. Independent linear process (15) on the AR kernels: ARTM-3 and FHTM-3. The models are evaluated on the MS-COCO (left) and PartiPrompts (right) with CLIPScore and PickScore every 10K training iterations across 100K iterations. Observe that on the AR kernels trained with the independent linear process are far superior to the ones trained with the dependent linear process.

## A.5 DTM Kernel expressiveness

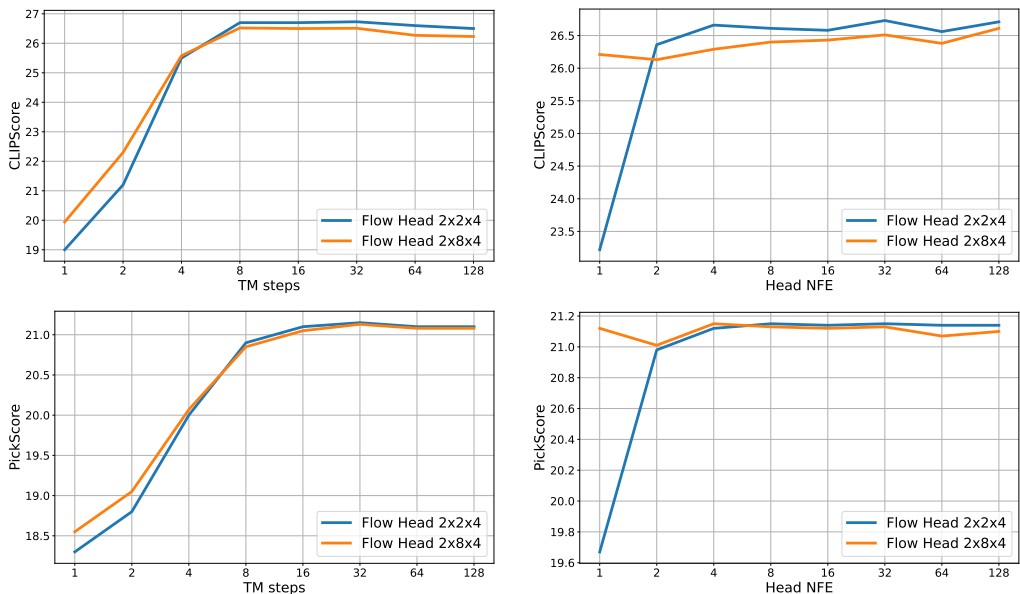

Figure 10: Impact of flow head patch size: $2 \times 2 \times 4$ vs. $2 \times 8 \times 4$, on the DTM performance, evaluated across varying numbers of TM steps (Left, with 32 Head NFE) and variying number of Head NFE (Right, with 32 TM steps). The metrics are CLIPScore (Top) and PickScore (Bottom) computed on the PartiPrompts dataset. On low number of TM steps, the larger flow head patch size shows an advantage in both metrics. On high number of TM steps, both patch sizes yield comparable results. This aligns with Theorem 1, which predicts that for infinitesimal steps size, the entries of $Y \in \mathbb{R}^d$ defined in equation 11 become independent.

## A.6 Scheduler ablation for independent linear process

We have experimented with two transition scheduler options: uniform (as described in 15) and "exponential", i.e., $\frac{t}{T} \in \{0, 0.5, 0.75, 1\}$. The results for ARTM and FHTM are reported in Table 12 and show almost the same performance with a slight benefit towards exponential in DiT architecture and these are used in our main implementations.

Table 12: Comparison of uniform and exponential transition steps.

| Kernel | Arch | TM Steps | Scheduler | MS-COCO | | PartiPrompts | |
| | | | | CLIPScore ↑ | PickScore ↑ | CLIPScore ↑ | PickScore ↑ |
|---|---|---|---|---|---|---|---|
| ARTM | DiT | 3 | Uniform | 26.0 | 20.8 | 26.8 | 20.8 |
| | | | Exponential | 26.1 | 20.9 | 27.0 | 20.9 |
| FHTM | DiT | 3 | Uniform | 25.9 | 21.0 | 26.9 | 20.9 |
| | | | Exponential | 26.1 | 21.0 | 27.0 | 20.9 |
| FHTM | LLM | 3 | Uniform | 26.1 | 21.0 | 27.0 | 21.0 |
| | | | Exponential | 26.1 | 21.1 | 27.0 | 21.0 |

## A.7 Additional generated images comparison

FM      MAR      FHTM      DTM

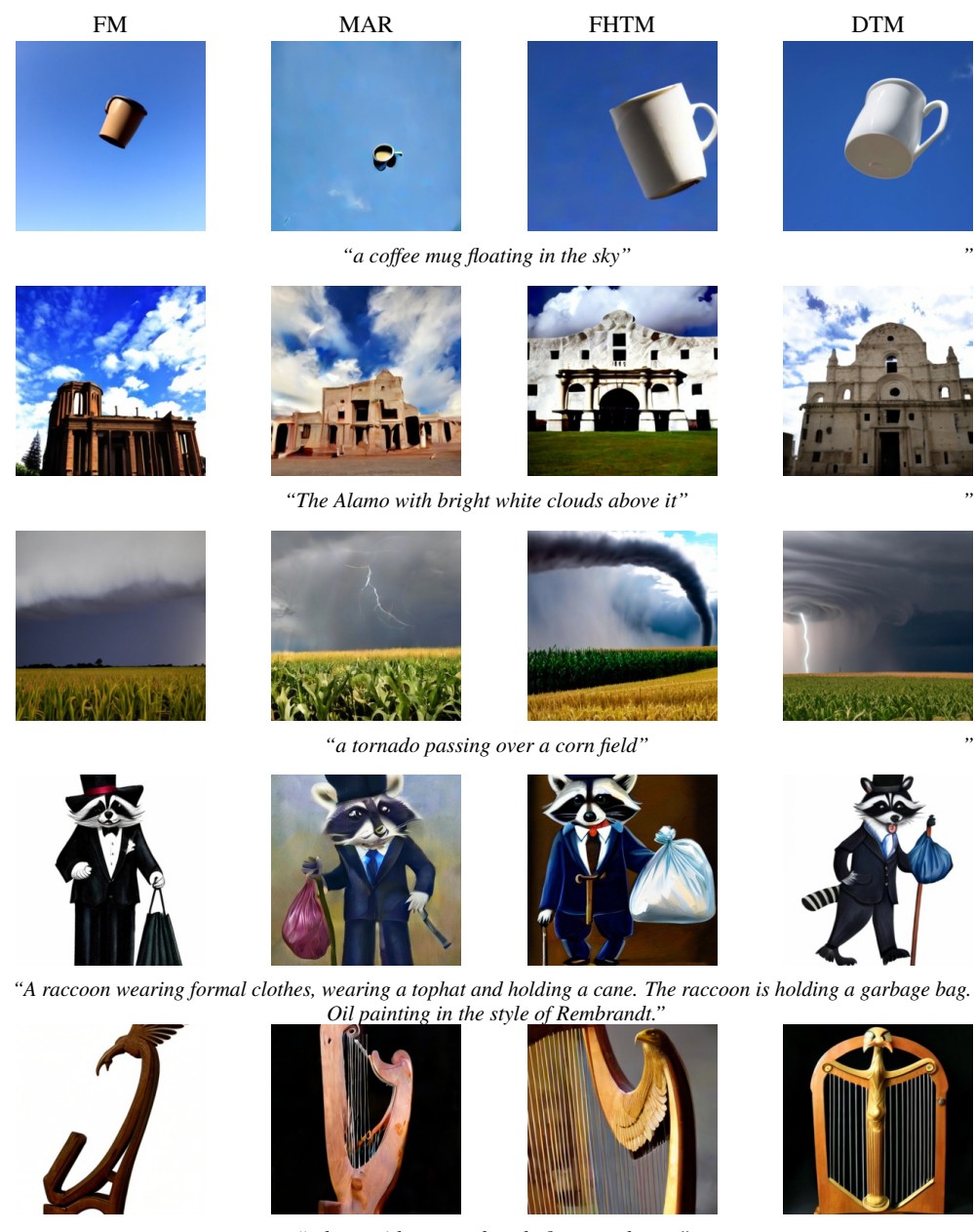

*"a coffee mug floating in the sky"*      *"*

*"The Alamo with bright white clouds above it"*      *"*

*"a tornado passing over a corn field"*      *"*

*"A raccoon wearing formal clothes, wearing a tophat and holding a cane. The raccoon is holding a garbage bag. Oil painting in the style of Rembrandt."*

*"a harp with a carved eagle figure at the top"*

Figure 11: Additional generated samples of FM, MAR, FHTM, and DTM with models that are trained for 1M iterations.

FM                    MAR                    FHTM                    DTM

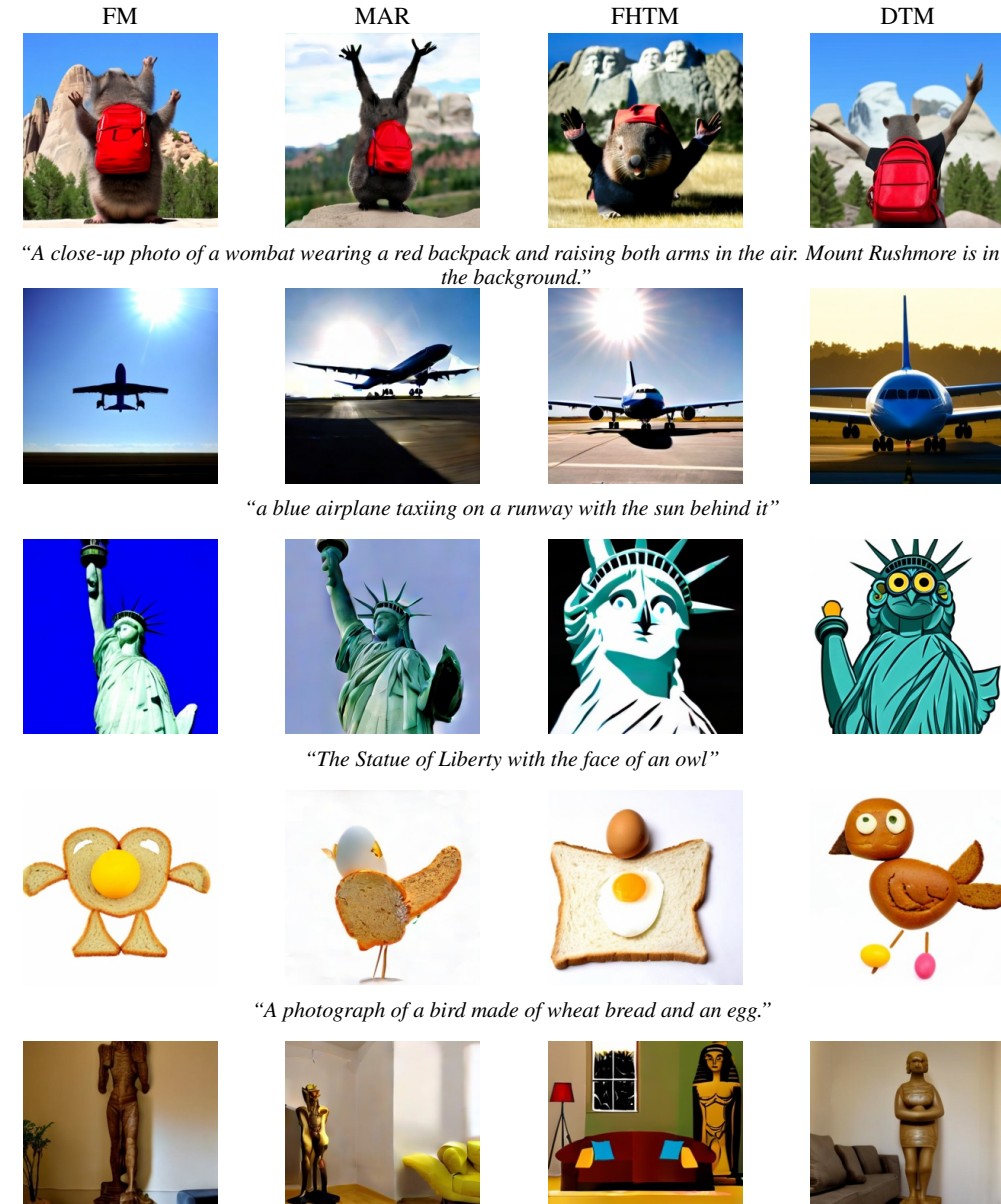

*"A close-up photo of a wombat wearing a red backpack and raising both arms in the air. Mount Rushmore is in the background."*

*"a blue airplane taxiing on a runway with the sun behind it"*

*"The Statue of Liberty with the face of an owl"*

*"A photograph of a bird made of wheat bread and an egg."*

*"a living room with a large Egyptian statue in the corner"*

Figure 12: Additional generated samples of FM, MAR, FHTM, and DTM with models that are trained for 1M iterations.

FM    MAR    FHTM    DTM

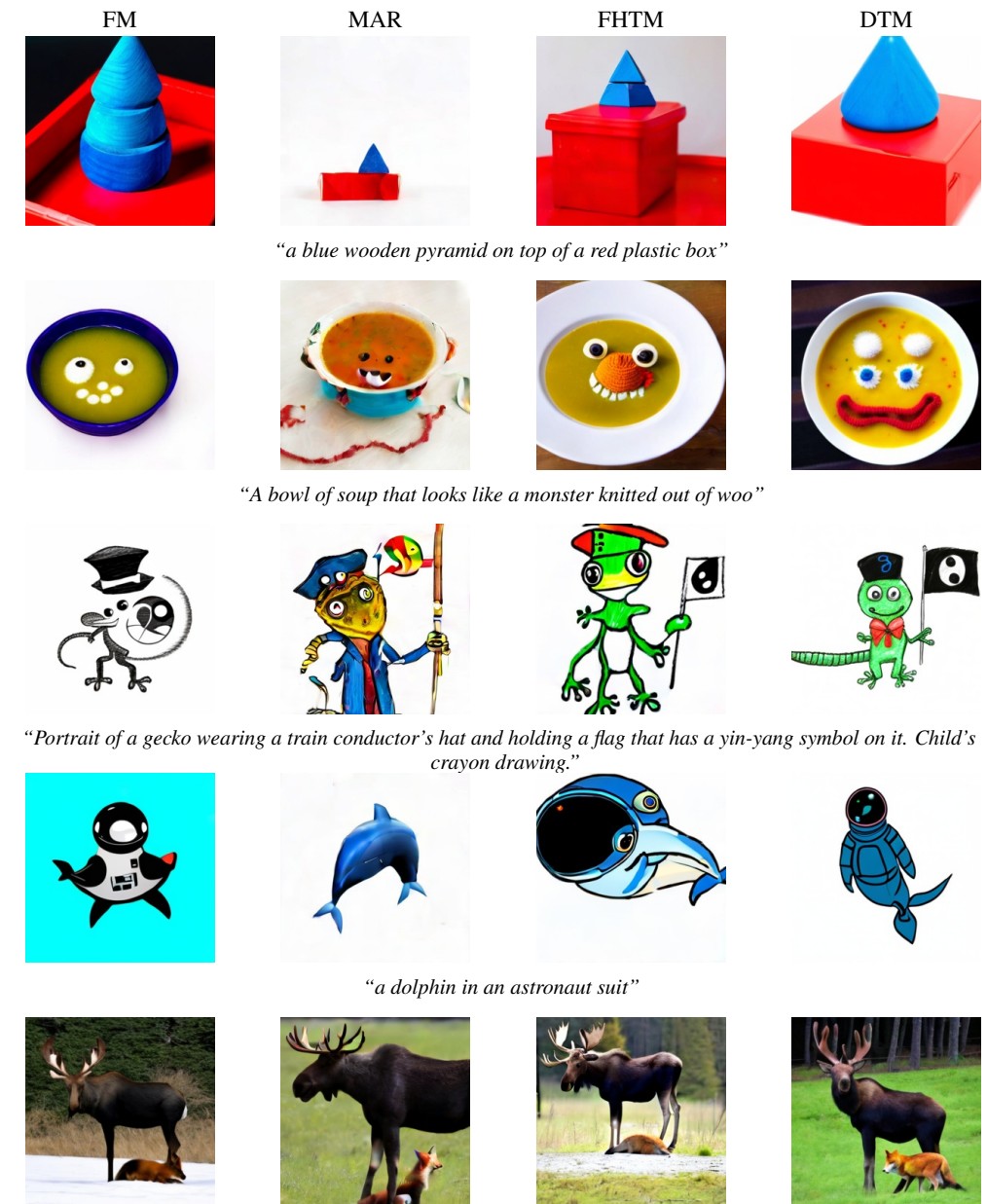

*"a blue wooden pyramid on top of a red plastic box"*

*"A bowl of soup that looks like a monster knitted out of woo"*

*"Portrait of a gecko wearing a train conductor's hat and holding a flag that has a yin-yang symbol on it. Child's crayon drawing."*

*"a dolphin in an astronaut suit"*

*"a moose standing over a fox"*

Figure 13: Additional generated samples of FM, MAR, FHTM, and DTM with models that are trained for 1M iterations.

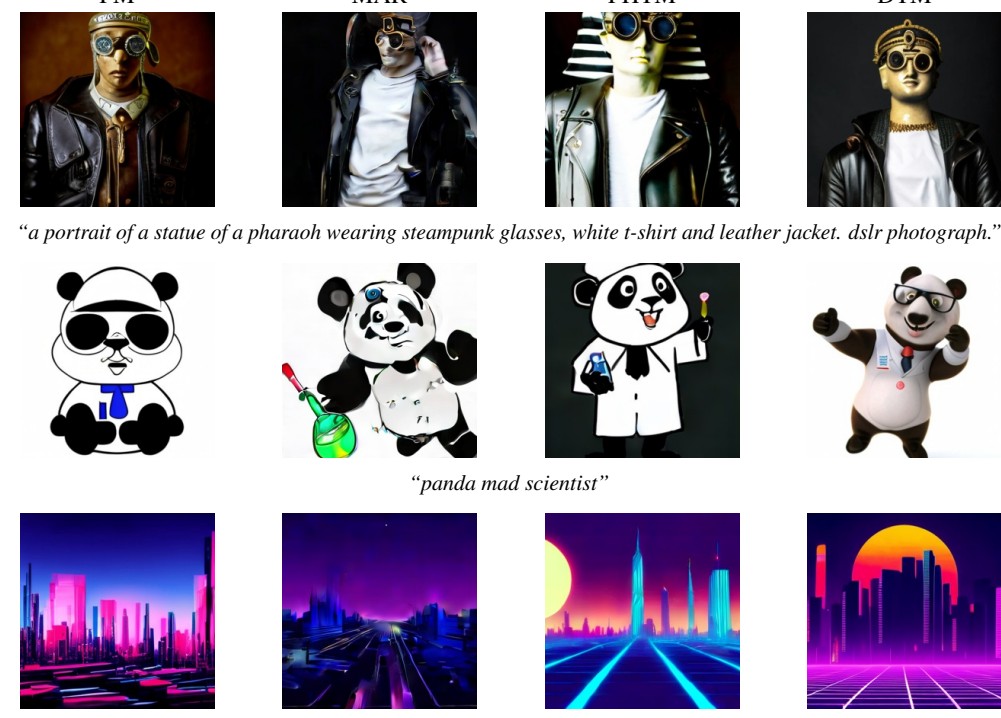

Figure 14: Additional generated samples of FM, MAR, FHTM, and DTM with models that are trained for 1M iterations.

| AR | ARTM-2 | ARTM-3 |
|---|---|---|

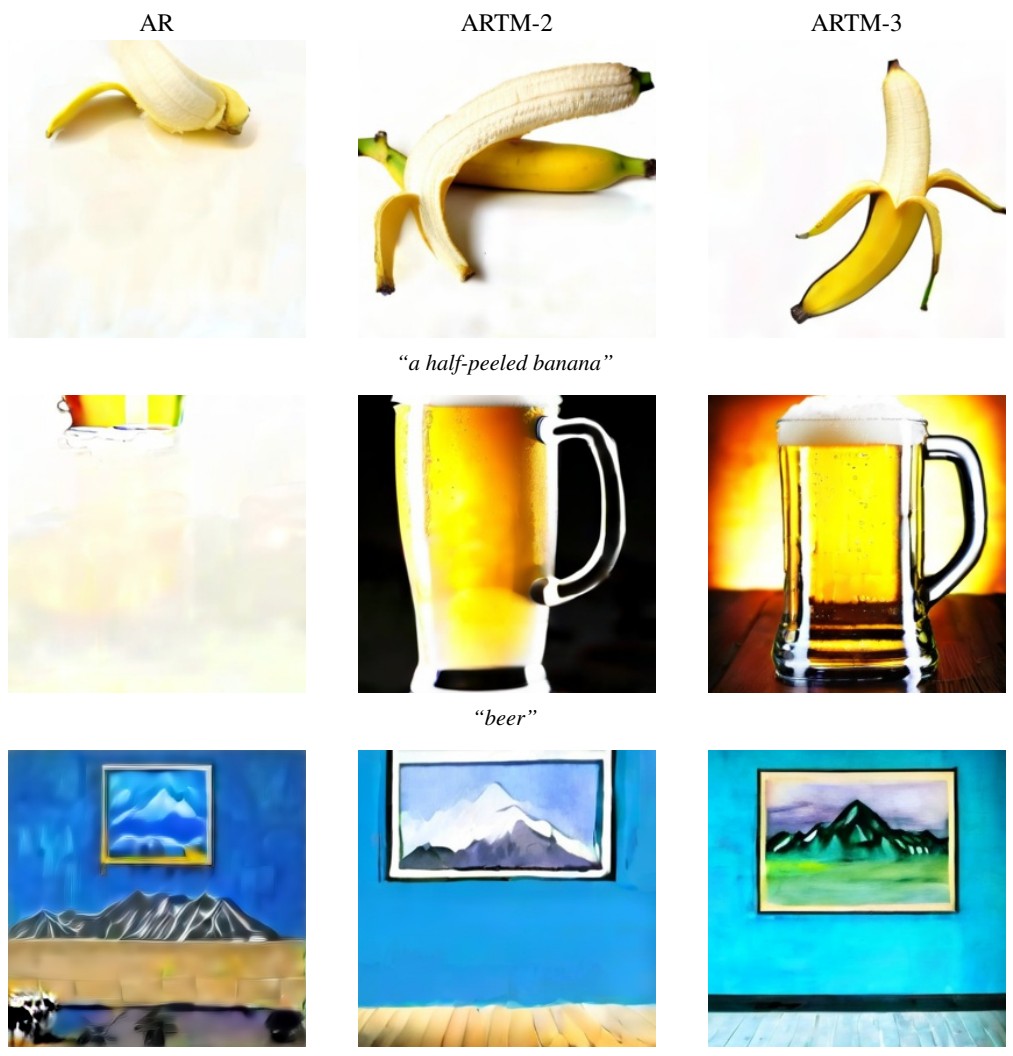

*"a half-peeled banana"*

*"beer"*

*"a blue wall with a large framed watercolor painting of a mountain"*

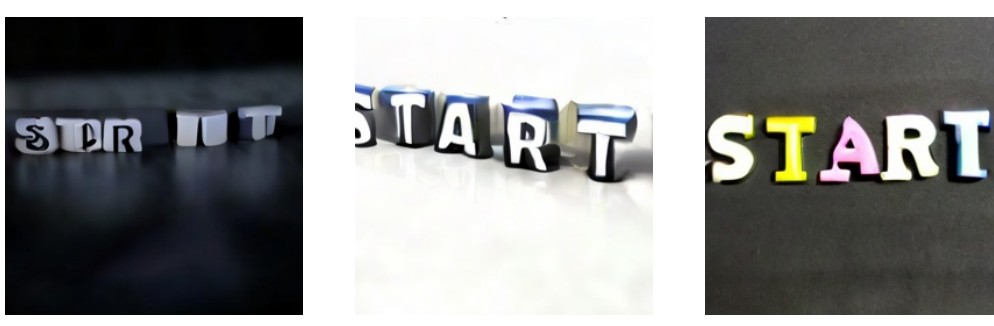

*"the word 'START' "*

Figure 15: Samples comparison of AR (left) vs. ARTM-2 (middle) vs. ARTM-3 (right) on models trained for 500K iteration with the DiT architecture.

## A.8 Generation process visualization

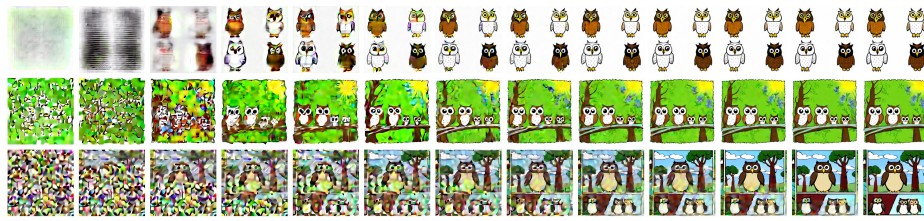

*"a comic about an owl family in the forest"*

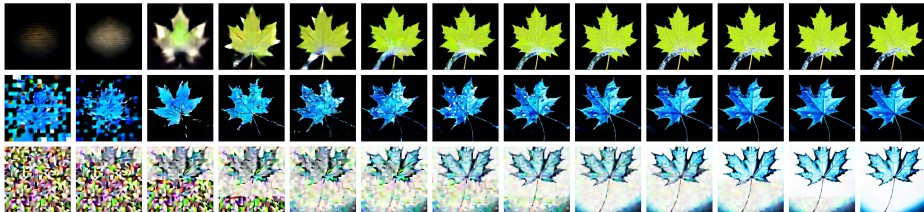

*"A photo of a maple leaf made of water."*

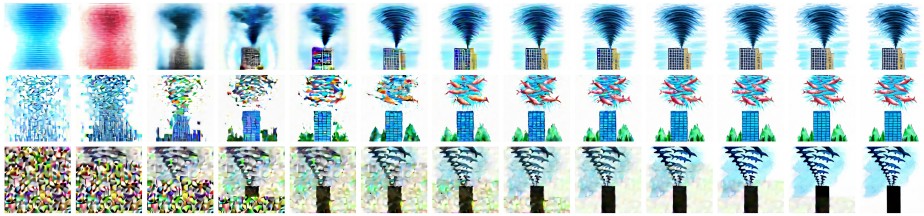

*"A tornado made of sharks crashing into a skyscraper. painting in the style of watercolor."*

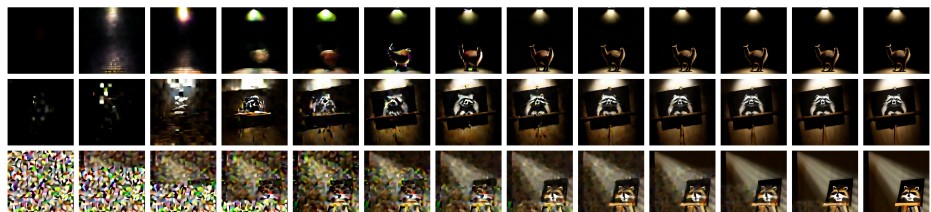

*"A single beam of light enter the room from the ceiling. The beam of light is illuminating an easel. On the easel there is a Rembrandt painting of a raccoon"*

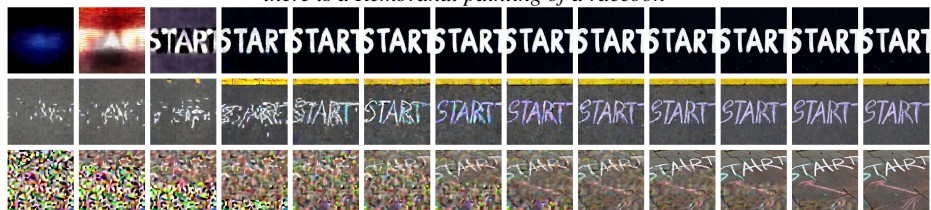

*"the word 'START' written in chalk on a sidewalk"*

Figure 16: Generation process of FM (first row), DTM (second row), and FHTM (third row) with models that are trained for 1M iterations. FM and DTM are visualized using a denoising estimation. FHTM-3 is visualized with 4 intermediates per transition step.

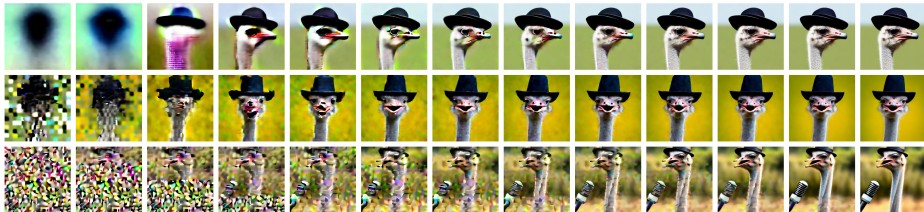

*"a photograph of an ostrich wearing a fedora and singing soulfully into a microphone"*

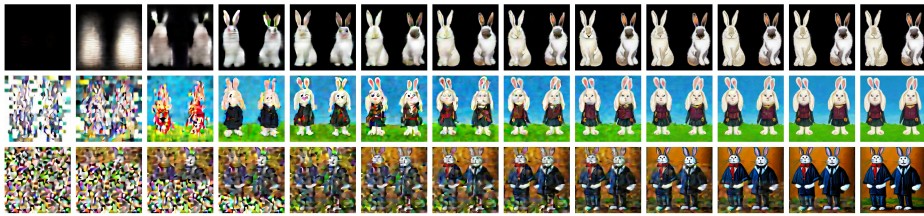

*"An oil painting of two rabbits in the style of American Gothic, wearing the same clothes as in the original."*

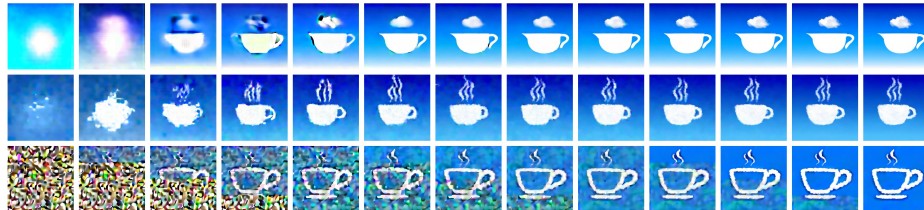

*"a cloud in the shape of a teacup"*

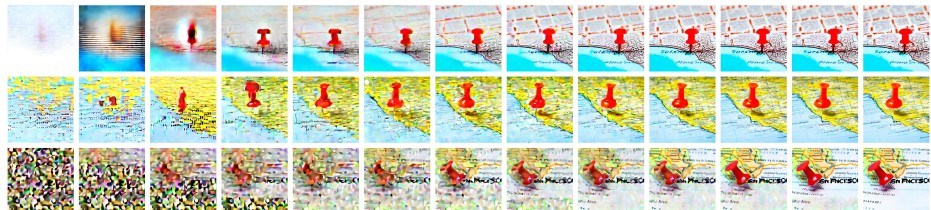

*"A map of the United States with a pin on San Francisco"*

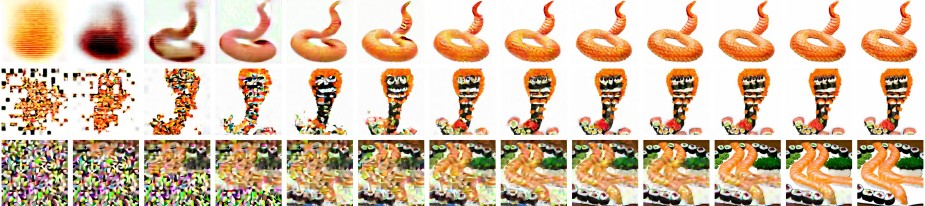

*"A giant cobra snake made from sushi"*

Figure 17: Generation process of FM (first row), DTM (second row), and FHTM (third row) with models that are trained for 1M iterations. FM and DTM are visualized using a denoising estimation. FHTM-3 is visualized with 4 intermediates per transition step.

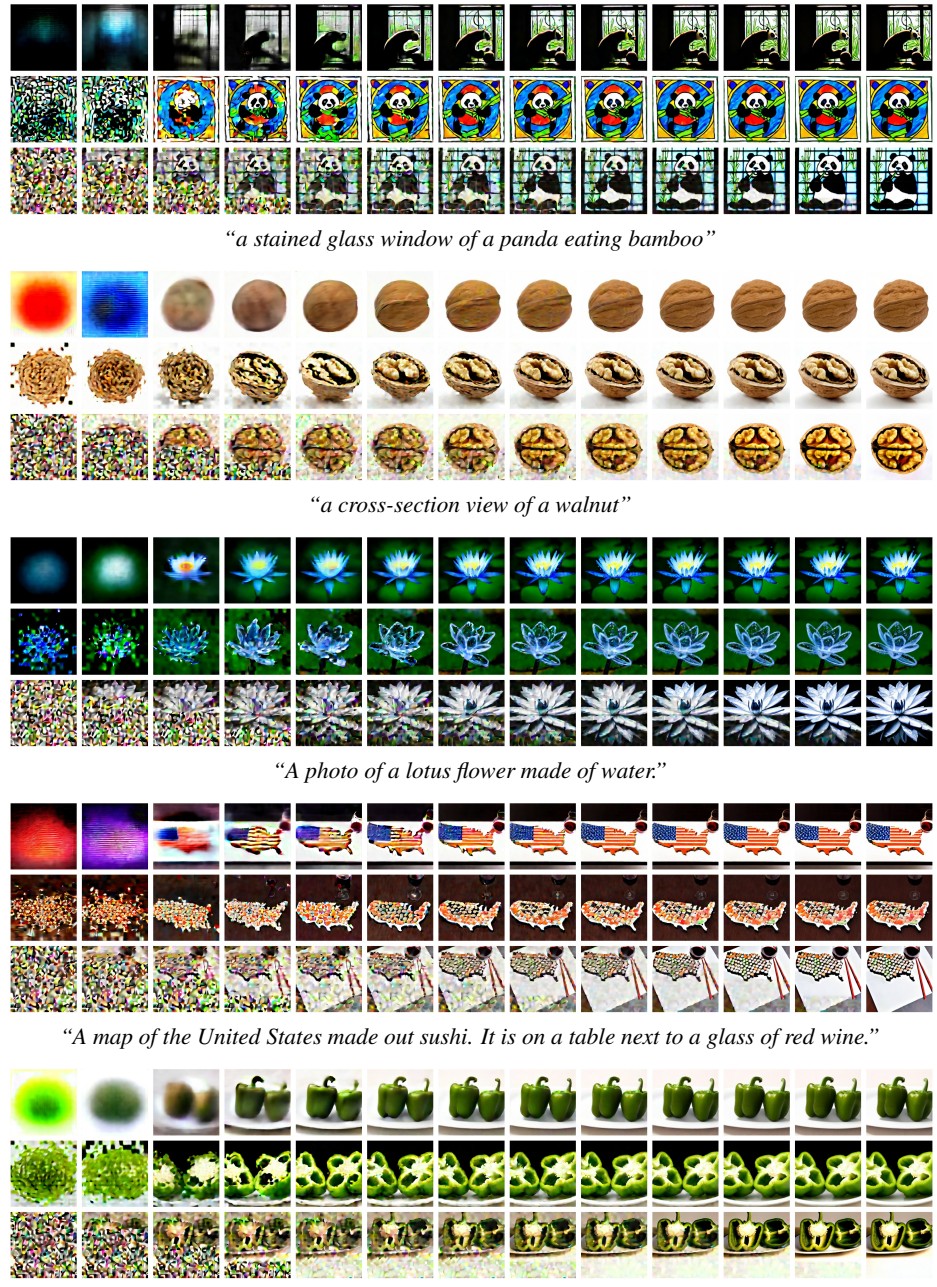

*"a stained glass window of a panda eating bamboo"*

*"a cross-section view of a walnut"*

*"A photo of a lotus flower made of water."*

*"A map of the United States made out sushi. It is on a table next to a glass of red wine."*

*"a green pepper cut in half on a plate"*

Figure 18: Generation process of FM (first row), DTM (second row), and FHTM (third row) with models that are trained for 1M iterations. FM and DTM are visualized using a denoising estimation. FHTM-3 is visualized with 4 intermediates per transition step.

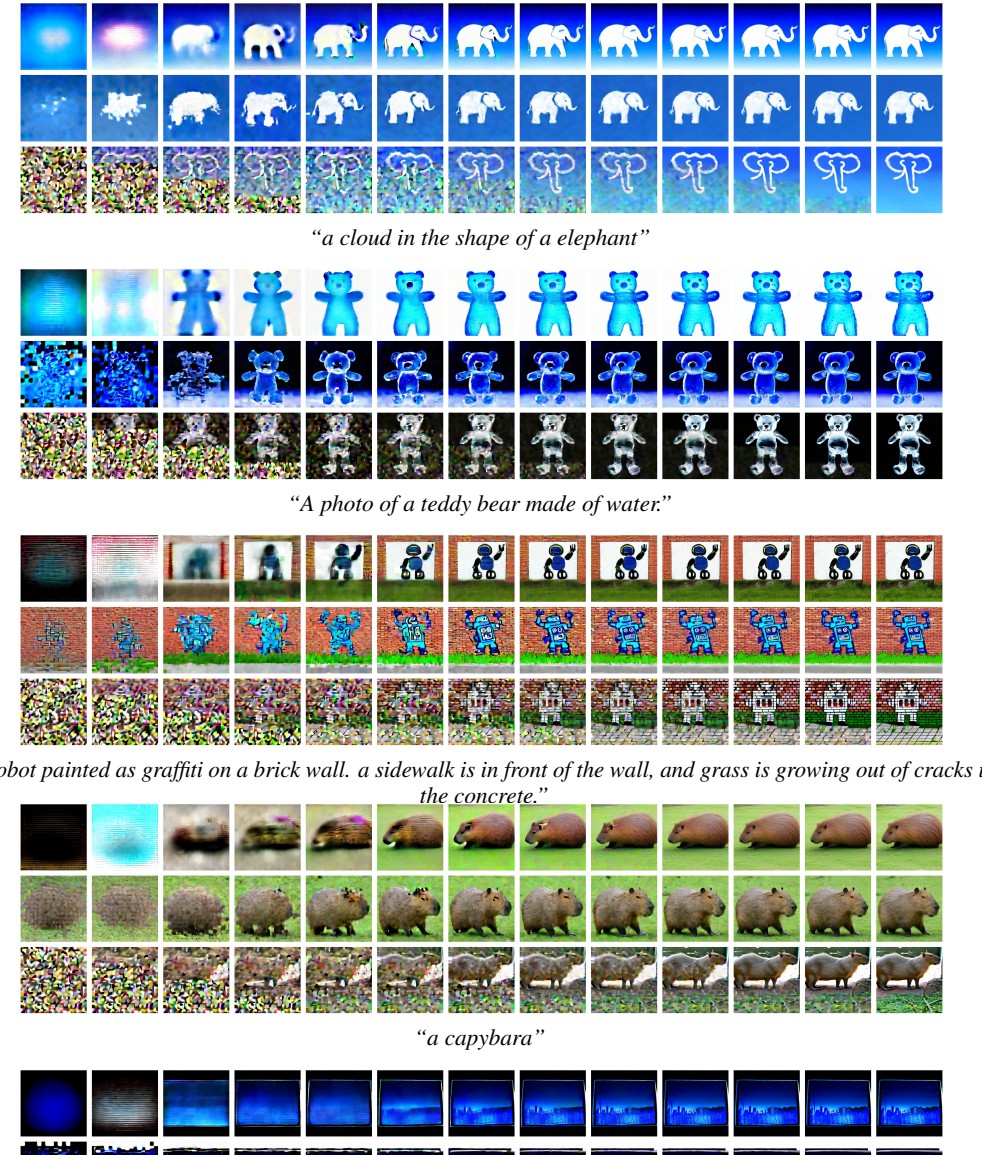

*"a cloud in the shape of a elephant"*

*"A photo of a teddy bear made of water."*

*"a robot painted as graffiti on a brick wall. a sidewalk is in front of the wall, and grass is growing out of cracks in the concrete."*

*"a capybara"*

*"A television made of water that displays an image of a cityscape at night."*

Figure 19: Generation process of FM (first row), DTM (second row), and FHTM (third row) with models that are trained for 1M iterations. FM and DTM are visualized using a denoising estimation. FHTM-3 is visualized with 4 intermediates per transition step.

## A.9   Classifier free guidance sensitivity

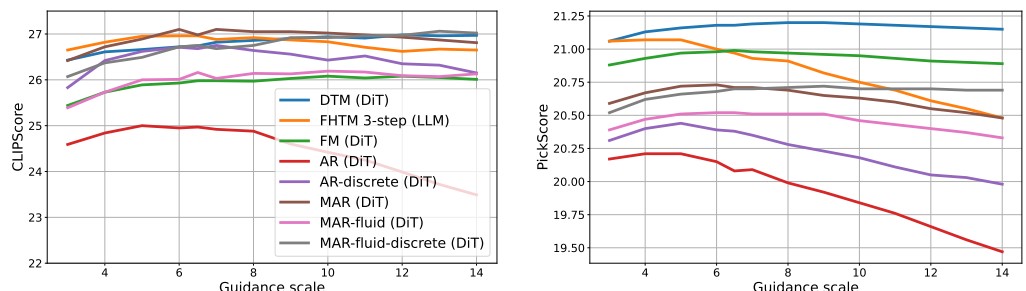

Figure 20: CLIPScore vs. CFG guidance scale (left) and PickScore vs. CFG guidance scale (right) of DTM and FHTM variants, and the baselines: FM, AR, AR-Discrete, MAR, MAR-Fluid, MAR-Fluid-Discrete on the PartiPrompts dataset.

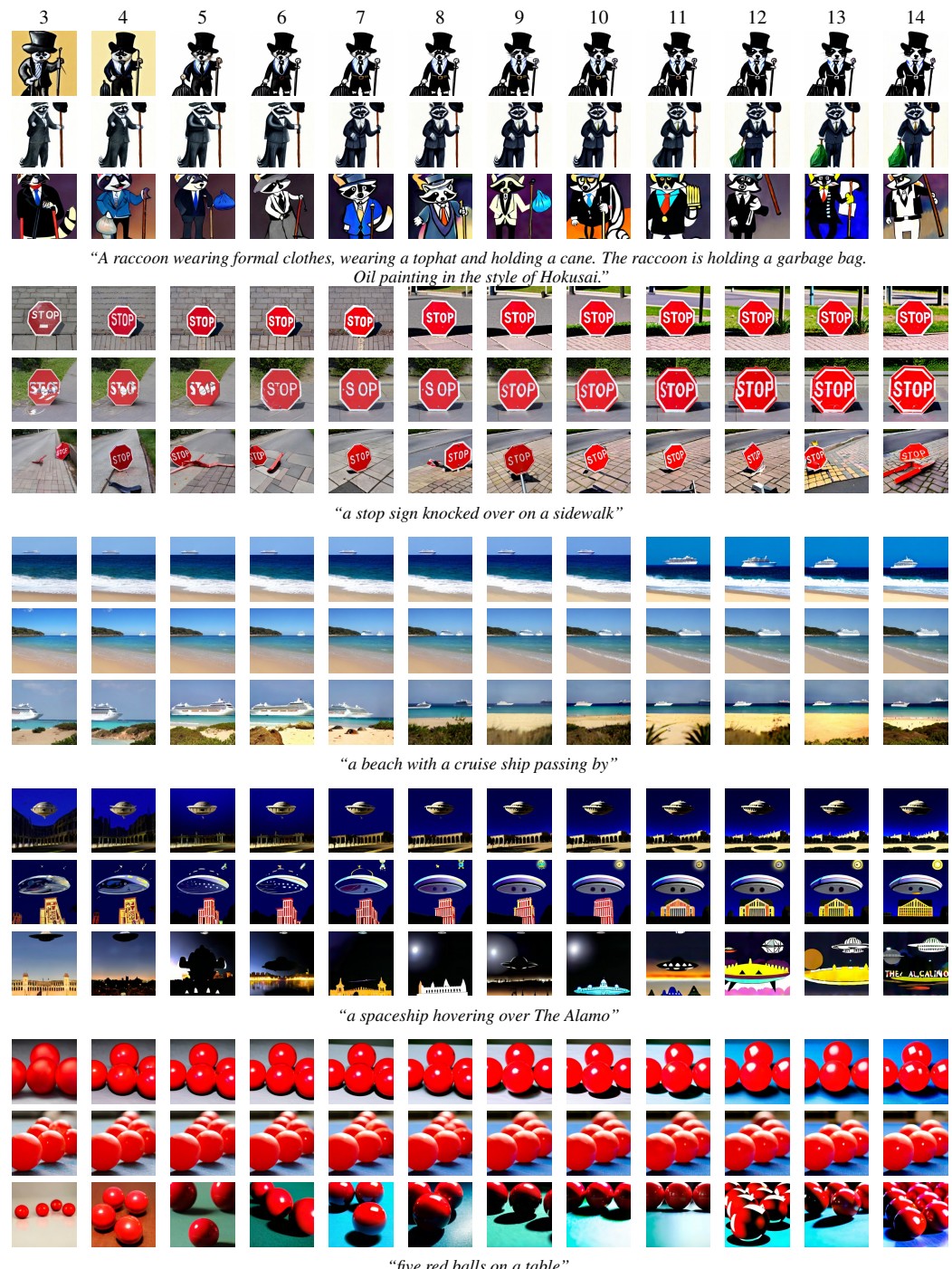

Figure 21: Classifier free guidance sensitivity for FM (first row), DTM (second row), and FHTM (third row) with models that are trained for 500k iterations.

# B  Training and sampling algorithms

Algorithms 1 and 2 describe and training and sampling (resp.) of transition matching for a general supervision process, kernel parametrization, and kernel modeling. In this section, we provide training and sampling algorithms tailored to the specific desgin choices of our three variants: (i) DTM is described in Figure 22, (ii) ARTM is described in Figure 23, and (iii) FHTM is described in Figure 24. Additionally, we provide Python code of a training step for each variant: (i) DTM in Figure 25, (ii) ARTM in Figure 26, and (iii) FHTM in Figure 27.

**Algorithm 3** DTM Training

**Require:** $p_T$      ▷ Data
**Require:** $T$      ▷ Number of TM steps
1: **while** not converged **do**
2:      Sample $X_T \sim p_T$
3:      Sample $t \sim \mathcal{U}([T-1])$

4:      Sample $X_0 \sim N(0, I_d)$
5:      $X_t \leftarrow \left(1 - \frac{t}{T}\right) X_0 + \frac{t}{T} X_T$        Sample $(X_t, Y) \sim q_{t,Y|T}(\cdot|X_T)$
6:      $Y \leftarrow X_T - X_0$

7:      $h_t \leftarrow f_t^\theta(X_t)$
8:      **parallel for** $i = 1, ..., n$ **do**
9:          Sample $Y_0^i \sim N(0, I_{d/n})$
10:        Sample $s \sim \mathcal{U}([0, 1])$        $\mathcal{L}(\theta) \leftarrow \hat{D}(Y, p_{Y|t}^\theta(\cdot|X_t))$
11:        $Y_s^i \leftarrow (1 - s) Y_0^i + s Y^i$
12:        $\mathcal{L}^i(\theta) \leftarrow \left\| g_{s,t}^\theta(Y_s^i, h_t^i) - \left(Y^i - Y_0^i\right) \right\|^2$
13:      **end for**
14:      $\mathcal{L}(\theta) \leftarrow \frac{1}{n} \sum_i \mathcal{L}^i(\theta)$

15:      $\theta \leftarrow \theta - \gamma \nabla_\theta \mathcal{L}$ ▷ Optimization step
16: **end while**
17: **return** $\theta$

---

**Algorithm 4** DTM Sampling

**Require:** $\theta$      ▷ Trained model
**Require:** $T$      ▷ Number of TM steps
1: Sample $X_0 \sim \mathcal{N}(0, I_d)$
2: **for** $t = 0$ **to** $T - 1$ **do**

3:      $h_t \leftarrow f^\theta(X_t, t)$
4:      **parallel for** $i = 1, ..., n$ **do**
5:        Sample $Y_0^i \sim N(0, I_{d/n})$        Sample $Y \sim p_{Y|t}^\theta(\cdot|X_t)$
6:        $Y^i \leftarrow \texttt{ode\_solve}\left(Y_0^i, g_{\cdot,t}^\theta\left(\cdot, h_t^i\right)\right)$
7:      **end for**

8:      $X_{t+1} \leftarrow X_t + \frac{1}{T} Y$        Sample $X_{t+1} \sim q_{t+1|t,Y}(\cdot|X_t, Y)$
9: **end for**
10: **return** $X_T$

---

Figure 22: $n$ is the effective sequence length after `patchify` layer. The *parallel for* operations run simultaneously across the "sequence length" dimension of the tensor; `ode_solve` is any generic ODE solver for solving equation 8.

---

**Algorithm 5** ARTM Training

---

**Require:** $p_T$ ▷ Data
**Require:** $T$ ▷ Number of TM steps
1: **while** not converged **do**
2:     Sample $X_T \sim p_T$
3:     Sample $t \sim \mathcal{U}([T-1])$

4:     Sample $X_{0,t} \sim N(0, I_d)$
5:     $X_t \leftarrow \left(1 - \frac{t}{T}\right) X_{0,t} + \frac{t}{T} X_T$
6:     Sample $X_{0,t+1} \sim N(0, I_d)$
7:     $X_{t+1} \leftarrow \left(1 - \frac{t+1}{T}\right) X_{0,t+1} + \frac{t+1}{T} X_T$       Sample $(X_t, Y) \sim q_{t,Y|T}(\cdot|X_T)$

8:     **parallel for** $i = 1, ..., n$ **do**
9:         $h_{t+1}^i \leftarrow f_t^\theta\left(X_t, X_{t+1}^{<i}\right)$
10:        Sample $Y_0^i \sim N(0, I_{d/n})$
11:        Sample $s \sim \mathcal{U}([0,1])$
12:        $Y_s^i \leftarrow (1-s) Y_0^i + s X_{t+1}^i$       $\mathcal{L}(\theta) \leftarrow \hat{D}(Y, p_{Y|t}^\theta(\cdot|X_t))$
13:        $\mathcal{L}^i(\theta) \leftarrow \left\| g_{s,t}^\theta(Y_s^i, h_{t+1}^i) - \left(X_{t+1}^i - Y_0^i\right)\right\|^2$
14:     **end for**
15:     $\mathcal{L}(\theta) \leftarrow \frac{1}{n} \sum_i \mathcal{L}^i(\theta)$

16:     $\theta \leftarrow \theta - \gamma \nabla_\theta \mathcal{L}$ ▷ Optimization step
17: **end while**
18: **return** $\theta$

---

---

**Algorithm 6** ARTM Sampling

---

**Require:** $\theta$ ▷ Trained model
**Require:** $T$ ▷ Number of TM steps
1: Sample $X_0 \sim \mathcal{N}(0, I_d)$
2: **for** $t = 0$ **to** $T - 1$ **do**
3:     **for** $i = 1, ..., n$ **do**
4:         $h_{t+1}^i \leftarrow f_t^\theta\left(X_t, X_{t+1}^{<i}\right)$
5:         Sample $Y_0^i \sim N(0, I_{d/n})$       Sample $X_{t+1} \sim p_{t+1|t}^\theta(\cdot|X_t)$
6:         $X_{t+1}^i \leftarrow \texttt{ode\_solve}\left(Y_0^i, g_{\cdot,t}^\theta\left(\cdot, h_{t+1}^i\right)\right)$
7:     **end for**
8: **end for**
9: **return** $X_T$

---

Figure 23: $n$ is the effective sequence length after `patchify` layer. The *parallel for* operations run simultaneously across the "sequence length" dimension of the tensor; `ode_solve` is any generic ODE solver for solving equation 8.

**Algorithm 7** FHTM Training

**Require:** $p_T$      ▷ Data
**Require:** $T$      ▷ Number of TM steps
1: **while** not converged **do**
2:     Sample $X_T \sim p_T$

3:     **parallel for** $t = 0, ..., T$ **do**
4:         Sample $X_{0,t} \sim N(0, I_d)$
5:         $X_t \leftarrow \left(1 - \frac{t}{T}\right) X_{0,t} + \frac{t}{T} X_T$     $\bigg\}$ Sample $(X_t, Y) \sim q_{t,Y|T}(\cdot|X_T)$
6:     **end for**

7:     **parallel for** $t = 0, ..., T-1, i = 1, ..., n$ **do**
8:         $h_{t+1}^i \leftarrow f_t^\theta \left(X_0, ..., X_t, X_{t+1}^{<i}\right)$
9:         Sample $Y_0^i \sim N(0, I_{d/n})$
10:       Sample $s \sim \mathcal{U}([0,1])$
11:       $Y_s^i \leftarrow (1-s) Y_0^i + s X_{t+1}^i$     $\bigg\}$ $\mathcal{L}(\theta) \leftarrow \hat{D}(Y, p_{Y|t}^\theta(\cdot|X_t))$
12:       $\mathcal{L}_t^i(\theta) \leftarrow \left\| g_s^\theta(Y_s^i, h_{t+1}^i) - \left(X_{t+1}^i - Y_0^i\right) \right\|^2$
13:     **end for**
14:     $\mathcal{L}(\theta) \leftarrow \frac{1}{nT} \sum_{i,t} \mathcal{L}_t^i(\theta)$

15:     $\theta \leftarrow \theta - \gamma \nabla_\theta \mathcal{L}$ ▷ Optimization step
16: **end while**
17: **return** $\theta$

---

**Algorithm 8** FHTM Sampling

**Require:** $\theta$      ▷ Trained model
**Require:** $T$      ▷ Number of TM steps
1: Sample $X_0 \sim \mathcal{N}(0, I_d)$
2: **for** $t = 0$ **to** $T-1$ **do**
3:     **for** $i = 1, ..., n$ **do**
4:         $h_{t+1}^i \leftarrow f_t^\theta \left(X_0, ..., X_t, X_{t+1}^{<i}\right)$
5:         Sample $Y_0^i \sim N(0, I_{d/n})$     $\bigg\}$ Sample $X_{t+1} \sim p_{t+1|t}^\theta(\cdot|X_t)$
6:         $X_{t+1}^i \leftarrow \texttt{ode\_solve}\left(Y_0^i, g_\cdot^\theta\left(\cdot, h_{t+1}^i\right)\right)$
7:     **end for**
8: **end for**
9: **return** $X_T$

---

Figure 24: $n$ is the effective sequence length after `patchify` layer. The *parallel for* operations run simultaneously across the "sequence length" dimension of the tensor; `ode_solve` is any generic ODE solver for solving equation 8.

```python
import torch
from torch import nn, Tensor
from einops import rearrange

def dtm_train_step(
        backbone:nn.Module,  # Denoted as `f^\theta`
        head:nn.Module,      # Denoted as `g^\theta`
        X_T:Tensor,          # Image from training set `X_T~p_T`
        T:int                # Number of TM steps
        patch_size:int       # Patch size
    ) -> Tensor:
    # Convert image to sequence using patchify
    X_T = rearrange(
        X_T,
        "b c (h dh) (w dw) -> b (h w) (dh dw c)",
        dh=patch_size,
        dw=patch_size,
    )
    bsz, seq_len = X_T.shape[:2]

    # Sample time step `t~U[T-1]`
    t = torch.randint(0, T, (bsz,))

    # Sample a pair `(X_t,Y)~q_{t,Y|T}(./X_T)``
    X_0 = torch.randn_like(X_T)
    X_t = (1-t/T).view(-1,1,1) * X_0 + (t/T).view(-1,1,1) * X_T
    Y = X_T - X_0

    # Backbone forward
    h_t = backbone(X_t, t)

    # Reshape sequence for head
    h_t = h_t.view(bsz*seq_len, -1)
    Y = Y.view(bsz*seq_len, -1)
    t = t.repeat_interleave(seq_len)

    # Flow matching loss with the head as velocity and Y as target
    Y_0 = torch.randn_like(Y)
    s = torch.rand(bsz*seq_len)
    Y_s = (1-s).view(-1,1) * Y_0 + s.view(-1,1) * Y

    # Head forward
    u = head(h_t, t, Y_s, s)
    loss = torch.nn.functional.mse_loss(u, Y - Y_0)

    return loss
```

Figure 25: Python code for DTM training

```python
1   import torch
2   from torch import nn, Tensor
3
4   def artm_train_step(
5           backbone:nn.Module,  # Denoted as `f^\theta`
6           head:nn.Module,      # Denoted as `g^\theta`
7           X_T:Tensor,          # Image from training set `X_T~p_T`
8           T:int                # Number of TM steps
9           patch_size:int       # Patch size
10      ) -> Tensor:
11      # Convert image to sequence using patchify
12      X_T = rearrange(
13          X_T,
14          "b c (h dh) (w dw) -> b (h w) (dh dw c)",
15          dh=patch_size,
16          dw=patch_size,
17      )
18      bsz, seq_len = X_T.shape[:2]
19
20      # Sample time step `t~U[T-1]`
21      t = torch.randint(0, T, (bsz,))
22
23      # Sample a pair `(X_t,Y)~q_{t,Y|T}(.|X_T)`
24      X_0_t = torch.randn_like(X_T)
25      X_t = (1-t/T).view(-1,1,1) * X_0_t +  (t/T).view(-1,1,1) * X_T
26      X_0_tp1 = torch.randn_like(X_T)
27      Y = (1-(t+1)/T).view(-1,1,1) * X_0_tp1 + ((t+1)/T).view(-1,1,1) * X_T
28
29      # Backbone forward
30      output = backbone(torch.cat([X_t, Y], dim=1), t)
31      h_tp1 = output[:, seq_len-1:-1]
32
33      # Reshape sequence for head
34      h_tp1 = h_tp1.view(bsz*seq_len, -1)
35      Y = Y.view(bsz*seq_len, -1)
36      t = t.repeat_interleave(seq_len)
37
38      # Flow matching loss with the head as velocity and Y as target
39      Y_0 = torch.randn_like(Y)
40      s = torch.rand(bsz*n_tokens)
41      Y_s = (1-s).view(-1,1) * Y_0 + s.view(-1,1) * Y
42
43      # Head forward
44      u = head(h_tp1, t, Y_s, s)
45      loss = torch.nn.functional.mse_loss(u, Y - Y_0)
46
47      return loss
```

Figure 26: Python code for ARTM training

```python
import torch
from torch import nn, Tensor

def fhtm_train_step(
        backbone:nn.Module,  # Denoted as `f^\theta`
        head:nn.Module,      # Denoted as `g^\theta`
        X_T:Tensor,          # Image from training set `X_T~p_T`
        T:int                # Number of TM steps
        patch_size:int       # Patch size
    ) -> Tensor:
    # Convert image to sequence using patchify
    X_T = rearrange(
        X_T,
        "b c (h dh) (w dw) -> b (h w) (dh dw c)",
        dh=patch_size,
        dw=patch_size,
    )

    bsz, seq_len, d = X_T.shape

    # Sample a pair `(X_t,Y)~q_{t,Y|T}(.|X_T)``
    boi = torch.zeros(bsz,1,d) # begin of image token
    X_FH = [boi]
    for t in range(1,T+1):
        X_0_t = torch.randn_like(X_T)
        X_FH.append(
            (1-t/T) * X_0_t + t/T * X_T
        )
    X_FH = torch.cat(X_FH, dim=1)
    X_t = X_FH[:, :-1]
    Y = X_FH[:, 1:]

    # forward for teacher forcing
    h_tp1 = backbone(X_t)

    # Reshape sequence for head
    h_tp1 = h_tp1.view(bsz*seq_len*T, -1)
    Y = Y.view(bsz*seq_len*T, -1)

    # Flow matching loss with the head as velocity and Y as target
    Y_0 = torch.randn_like(Y)
    s = torch.rand(bsz*seq_len*T)
    Y_s = (1-s).view(-1,1) * Y_0 + s.view(-1,1) * Y

    # Head forward
    u = head(h_tp1, Y_s, s)
    loss = torch.nn.functional.mse_loss(u, Y - Y_0)

    return loss
```

Figure 27: Python code for FHTM training

## C Convergence of DTM to flow matching

Here we want to prove the following fact: Assume we have a sequence of Markov chains $\{X_0, X_h, X_{2h}, \ldots, X_1\}$, with an initial state $X_0 = x$, where $h = \frac{1}{T}$ and $T \to \infty$. For convenience note that we index the Markov states with fractions $\ell h$, $\ell \in [T]$, and we denote the RV

$$Y_t = \frac{X_{t+h} - X_t}{h}. \tag{19}$$

Assume the Markov chains satisfy:

1. The function $f_t(x) = \mathbb{E}[Y_t | X_t = x]$ is Lipshcitz continuous. By Lipschitz we mean that $\|f_s(y) - f_t(x)\| \leq c_L (|s - t| + \|x - y\|)$.

2. For $\ell \in [k]$ the quadratic variation satisfies, $\mathbb{E}\left[\|Y_{\ell h}\|^2 | X_0 = x\right] \leq c(x)$.

Let $k = k(h) \in \mathbb{N}$ be an integer-valued function of $h$ such that $k \to \infty$ and $\frac{1}{2} \geq kh \to 0$ as $h \to 0$. We will prove that the random variable

$$\frac{X_{kh} - X_0}{kh} \tag{20}$$

*converges in mean* to $f_0(x)$. That is, we want to show

**Theorem 2.** *Considering a sequence of Markov processes $\{X_0, X_h, X_{2h}, \ldots, X_1\}$ satisfying the assumptions above, then*

$$\lim_{h \to 0} \mathbb{E}\left[\left\|\frac{X_{kh} - X_0}{kh} - f_0(X_0)\right\|^2 \bigg| X_0 = x\right] = 0. \tag{21}$$

*Proof.* First,

$$\mathbb{E}\left[\left\|\frac{X_{kh} - X_0}{kh} - f_0(X_0)\right\|^2 \bigg| X_0 = x\right] = \mathbb{E}\left[\left\|\frac{1}{k}\sum_{\ell=0}^{k-1} Y_{\ell h} - f_0(X_0)\right\|^2 \bigg| X_0 = x\right] \tag{22}$$

and if we open the squared norm we get three terms:

$$\mathbb{E}\left[\|f_0(X_0)\|^2 \bigg| X_0 = x\right] = \|f_0(x)\|^2. \tag{23}$$

$$\mathbb{E}\left[\frac{1}{k}\sum_{\ell=0}^{k-1} Y_{\ell h} \cdot f_0(X_0) \bigg| X_0 = x\right] = f_0(x) \cdot \frac{1}{k}\sum_{\ell=0}^{k-1} \mathbb{E}\left[Y_{\ell h} \bigg| X_0 = x\right] \tag{24}$$

$$\mathbb{E}\left[\frac{1}{k^2}\sum_{\ell=0}^{k-1}\sum_{m=0}^{k-1} Y_{\ell h} \cdot Y_{mh} \bigg| X_0 = x\right] = \frac{1}{k^2}\sum_{\ell=0}^{k-1} \mathbb{E}\left[\|Y_{\ell h}\|^2 | X_0 = x\right]$$

$$+ \frac{2}{k^2}\sum_{\ell=0}^{k-1}\sum_{m=0}^{\ell-1} \mathbb{E}\left[Y_{\ell h} \cdot Y_{mh} \bigg| X_0 = x\right]$$

We will later show that $\mathbb{E}[Y_{\ell h} | X_0 = x] = f_0(x) + O(kh)$ and for $\ell \neq m$ we have $\mathbb{E}[Y_{\ell h} \cdot Y_{mh} | X_0 = x] = \|f_0(x)\|^2 + O(kh)$. Plugging these we get that equation 22 equals

$$\|f_0(x)\|^2 - 2\|f_0(x)\|^2 + \frac{k^2 - k}{k^2}\|f_0(x)\|^2 + O(kh + k^{-1}) \to 0, \tag{25}$$

as $h \to 0$, where we used assumption 2 above to bound $\mathbb{E}\left[\|Y_{\ell h}\|^2 \,|X_0 = x\right] \leq c(x)$. Now to conclude we show

$$\|\mathbb{E}\left[Y_{\ell h}|X_0 = x\right] - f_0(x)\| = \|\mathbb{E}\left[\mathbb{E}\left[Y_{\ell h}|X_{\ell h}\right]|X_0 = x\right] - f_0(x)\| \tag{26}$$
$$= \|\mathbb{E}\left[f_{\ell h}(X_{\ell h})|X_0 = x\right] - f_0(x)\| \tag{27}$$
$$= \|\mathbb{E}\left[f_{\ell h}(X_{\ell h}) - f_0(X_0)|X_0 = x\right]\| \tag{28}$$
$$= \mathbb{E}\left[\|f_{\ell h}(X_{\ell h}) - f_0(X_0)\| \,|X_0 = x\right] \tag{29}$$
$$\leq c_L \mathbb{E}\left[\ell h + \|X_{\ell h} - X_0\| \,|X_0 = x\right] \tag{30}$$
$$\leq c_L \mathbb{E}\left[\ell h + \sum_{m=0}^{\ell-1}\|X_{(m+1)h} - X_{mh}\| \,\Big|X_0 = x\right] \tag{31}$$
$$\leq O(kh) + c_L h \sum_{m=0}^{\ell-1}\mathbb{E}\left[\|Y_{mh}\| \,\Big|X_0 = x\right] \tag{32}$$
$$\leq O(kh) + c_L h k \sqrt{c(x)} \tag{33}$$
$$= O(hk). \tag{34}$$

Now for $m < \ell$ we have

$$|\mathbb{E}\left[Y_{\ell h} \cdot Y_{mh} \mid X_0 = x\right] - f_0(x) \cdot \mathbb{E}\left[Y_{mh}|X_0 = x\right]| \tag{35}$$
$$= |\mathbb{E}\left[Y_{mh} \cdot (Y_{\ell h} - f_0(X_0)) \mid X_0 = x\right]| \tag{36}$$
$$= |\mathbb{E}\left[Y_{mh} \cdot \mathbb{E}\left[Y_{\ell h} - f_0(X_0)|X_{\ell h}\right] \mid X_0 = x\right]| \tag{37}$$
$$= |\mathbb{E}\left[Y_{mh} \cdot (f_{\ell h}(X_{\ell h}) - f_0(X_0)) \mid X_0 = x\right]| \tag{38}$$
$$\leq \mathbb{E}\left[|Y_{mh} \cdot (f_{\ell h}(X_{\ell h}) - f_0(X_0))| \mid X_0 = x\right] \tag{39}$$
$$\leq \mathbb{E}\left[\|Y_{mh}\| \|f_{\ell h}(X_{\ell h}) - f_0(X_0)\| \mid X_0 = x\right] \tag{40}$$
$$\leq \mathbb{E}\left[\|Y_{mh}\| c_l(kh + \sum_{j=0}^{\ell-1}\|X_{(j+1)h} - X_{jh}\| \mid X_0 = x\right] \tag{41}$$
$$\leq \mathbb{E}\left[\|Y_{mh}\| c_l(kh + h\sum_{j=0}^{\ell-1}\|Y_{jh}\|) \mid X_0 = x\right] \tag{42}$$
$$\leq O(kh) + c_L h \mathbb{E}\left[\sum_{j=0}^{\ell-1}\|Y_{mh}\| \|Y_{jh}\| \mid X_0 = x\right] \tag{43}$$
$$\leq O(kh) + \frac{c_L h}{2}\mathbb{E}\left[\sum_{j=0}^{\ell-1}\|Y_{mh}\|^2 + \|Y_{jh}\|^2 \mid X_0 = x\right] \tag{44}$$
$$= O(kh). \tag{45}$$

Therefore,

$$|\mathbb{E}\left[Y_{\ell h} \cdot Y_{mh} \mid X_0 = x\right] - f_0(x) \cdot f_0(x)| \tag{46}$$
$$\leq |\mathbb{E}\left[Y_{\ell h} \cdot Y_{mh} \mid X_0 = x\right] - f_0(x) \cdot \mathbb{E}\left[Y_{mh}|X_0 = x\right]| \tag{47}$$
$$+ |f_0(x) \cdot \mathbb{E}\left[Y_{mh}|X_0 = x\right] - f_0(x) \cdot f_0(x)| \tag{48}$$
$$\leq O(kh), \tag{49}$$

where we used equation 34 and equation 45, and the proof is done since $kh \to 0$ as $h \to 0$. $\qquad\square$

## C.1 The DTM case

We note show that the DTM process satisfies the two assumptions above. We recall that the DTM process is defined by $Y_t \sim q_{Y|t}(\cdot|X_t)$ where $Y = X_1 - X_0$.

First we check the Lipchitz property.

$$f_t(x) = \mathbb{E}\left[Y_t|X_t = x\right] \tag{50}$$

$$= \mathbb{E}\left[X_1 - X_0|X_t = x\right] \tag{51}$$

$$= u_t(x) \tag{52}$$

$$= \int \frac{x_1 - x}{1 - t} p_{1|t}(x_1|x)\mathrm{d}x_1 \tag{53}$$

$$= \int \frac{x_1 - x}{1 - t} \frac{p_{t|1}(x|x_1)p_1(x_1)}{\int p_{t|1}(x|x_1')p_1(x_1')\mathrm{d}x_1'}\mathrm{d}x_1 \tag{54}$$

which is Lipschitz for $t < 1$ as long as $p_{t|1}(x|x_1) > 0$ for all $x$, and is continuously differentiable in $t$ and $x$, both hold for the Gaussian kernel $p_{t|1}(x|x_1) = \mathcal{N}(x|tx_1, (1-t)I)$.

Let us check the second property. For this end we make the realistic assumption that our data is bounded, i.e., $\|X_1\| \le r$ for some constant $r > 0$. Then, consider some RV $X_1' - X_0' = Y_t \sim p_{Y|t}(\cdot|X_t)$. Then by definition we have that $X_{t+h} = X_t + h(X_1' - X_0')$ and $X_t = (1-t)X_0' + tX_1'$. Therefore,

$$\|X_{t+h}\| = \|X_t + h(X_1' - X_0')\| \tag{55}$$

$$= \left\|\frac{(1 - t - h)X_t + h(1 + t)X_1'}{(1 - t)}\right\| \tag{56}$$

$$\le \frac{(1 - (t + h))}{(1 - t)}\|X_t\| + h\frac{1 + t}{1 - t}r. \tag{57}$$

We apply this to $t + h = \ell h$ where $\ell \in [k]$ and therefore

$$\|X_{\ell h}\| \le \|X_{(\ell-1)h}\| + h\frac{1 + kh}{1 - kh}r \tag{58}$$

$$\le \|X_0\| + kh\frac{1 + kh}{1 - kh}r \tag{59}$$

$$\le \|x\| + \frac{3r}{2} = \tilde{c}(x) \tag{60}$$

where we used $kh \le \frac{1}{2}$. Finally,

$$\mathbb{E}\left[\|Y_t\|^2 |X_0 = x\right] = \mathbb{E}\left[\mathbb{E}\left[\|Y_t\|^2 |X_t\right]|X_0 = x\right] \tag{61}$$

$$= \mathbb{E}\left[\mathbb{E}\left[\|X_1' - X_0'\|^2 |X_t\right]|X_0 = x\right] \tag{62}$$

$$\le \mathbb{E}\left[\mathbb{E}\left[\left\|X_1' - \frac{X_t - tX_1'}{(1 - t)}\right\|^2 |X_t\right]|X_0 = x\right] \tag{63}$$

$$\le \mathbb{E}\left[\mathbb{E}\left[\left\|\frac{X_t - (1 + t)X_1'}{(1 - t)}\right\|^2 |X_t\right]|X_0 = x\right] \tag{64}$$

$$\le 2\mathbb{E}\left[\frac{1}{(1 - t)^2}\|X_t\|^2 + \frac{(1 + t)^2}{(1 - t)^2}r^2|X_0 = x\right], \tag{65}$$

where we used again $X_t = (1 - t)X_0' + tX_1'$. Lastly, applying this to $t = \ell h \le kh \le \frac{1}{2}$ and using equation 60 we get

$$\mathbb{E}\left[\|Y_{k\ell}\|^2 |X_0 = x\right] \le \frac{2}{(1 - t)^2}\tilde{c}(x)^2 + 2\frac{(1 + t)^2}{(1 - t)^2}r^2 = c(x). \tag{66}$$

