# OpenReview forum: "Transition Matching: Scalable and Flexible Generative Modeling"
_NeurIPS.cc/2025/Conference — NeurIPS 2025 poster_

### Official Review · Reviewer_kPcj · 2025-06-25

**Clarity:** 1
**Significance:** 2
**Originality:** 1
**Rating:** 4
**Confidence:** 4

**Summary:**

This paper introduces a general framework called Transition Matching (TM), which unifies continuous-state autoregressive models. The framework is built upon three core design choices: the supervision process, kernel parameterization, and the modeling paradigm. By making specific selections within these dimensions, the authors propose three powerful new variants: DTM, ARTM, and FHTM. The empirical results show that the proposed method FHTM can match the quality of full-sequence flow model in image generation.

**Questions:**

Please address the weaknesses above.

**Ethical Concerns:**

["NO or VERY MINOR ethics concerns only"]

**Final Justification:**

The paper presents a new parameterization that benefits from larger mode coverage.
Additionally, the comprehensive results show promising results for AR-diffusion models and provide apple-to-apple comparison results which can benefit the future researches. I recommend this paper to be accepted in NeurIPS 2025.

**Limitations:**

Yes.

**Paper Formatting Concerns:**

None.

**Quality:**

2

**Strengths And Weaknesses:**

**[Distinctions to previous works]**

The paper proposes a general framework of continuous state autoregressive models by defining the three major design spaces.

However, the overall writing was not easy to follow with unconventional notations and missing justifications and explanations.
First, design space analysis has been widely explored in previous works [1] (diffusion), [2-3] (diffusion and flow), [4] (diffusion, flow, and
CTMC). For me Sec. 2.1 seems as a preliminary, not a core component of the paper, and the author's claims of unifying flow, diffusion under TM is too bold.

In Sec. 2.2, the paper specifically focuses on continuous state autoregressive models, which I could not spot the differences of DTM from MAR [5]. While the authors propose modeling the latent difference, rather than the conventional $\epsilon$-prediction in diffusion models or velocity prediction in flow models, I am not convinced why this is needed and such a design choice should give better results. Similarly, the use of independent linear process is not justified. What does it exactly mean by a better regularity of the conditional reverse kernel?


**[DTM results]**

The authors show that DTM achieves the best performances to full sequence denoising such as FM.
However, I find the results contradictory since the DTM also outperforms AR-based methods that adopt causal attention.
Additionally, how does DTM compete against FM in inference time or training stability?


Minor typo: well-explored (L2), mixed use of discrete, continuous time (L125-126).



[1] Denoising Diffusion Implicit Models

[2] Flow Matching for Generative Modeling

[3] Stochastic Interpolants: A Unifying Framework for Flows and Diffusions

[4] Generator Matching: Generative modeling with arbitrary Markov processes

[5] Autoregressive Image Generation without Vector Quantization

---

> ### Author Rebuttal · Authors · 2025-07-30
>
> > The overall writing was not easy to follow with unconventional notations and missing justifications and explanations.
>
> We thank the reviewer for the feedback and are sorry to hear that parts of the paper were difficult to follow. We made an honest effort to find a notation to describe the design space of TM. If the reviewer can provide more specifics on what is unclear we will try and clarify further.
>
>
> > Design space analysis has been widely explored in previous works [1] (diffusion), [2-3] (diffusion and flow), [4] (diffusion, flow, and CTMC). For me Sec. 2.1 seems as a preliminary, not a core component of the paper.
>
> The reviewer correctly notes that previous works have explored learning transition kernels of Markov chains with a simulation free loss similar to equations 4 and 5, particularly in the context of diffusion and flow-based models.
> However, we identify several key differences that make Section 2.1 more than just a preliminary: (1) [1-4] focus on factorized transition kernels (e.g., Gaussian/deterministic kernels), with the exception of MAR [5] which is patch-wise factorized, and we further elaborate on DTM vs. MAR to an answer below. In contrast, we formulate a general, simulation-free learning methodology of **arbitrary** transition kernels. (2) we provide a formulation that allows arbitrary supervision processes and a systematic way to consider different kernel parameterizations; and (3) we feel the level of generality we chose allowed us to provide a full exposition to TM, including all design choices, within (hopefully clear) 2 pages. We expect this to allow future researchers to rather quickly get familiar with the field in a rather general manner, capturing the main design choices of existing methods.
>
>
> > The author's claims of unifying flow, diffusion under TM is too bold.
>
> TM includes diffusion/flow matching and AR, however we do not claim it to be the first framework to include both diffusion and flow matching (or AR to that matter). If the reviewer can point to where such a claim is made in the paper we are happy to clarify it in the text.
>
> > In Sec. 2.2, the paper specifically focuses on continuous state autoregressive models, which I could not spot the differences of DTM from MAR [5].
>
> DTM and MAR indeed both use the same _Modeling_, i.e.,  DiT backbone with a small flow/diffusion head but they are distinct in their _Supervising Process_ and _Parametrization_ which lead to different training and generation processes and notably lead to very different performance where MAR is **considerably sub-par** to DTM .
>
> In detail:
> Supervising Process:
> * DTM: $X_t = \left( 1-\frac{t}{T} \right)X_0 + \frac{t}{T}X_T$, where $X_0\sim\mathcal{N}(0, I)$.
>
> * MAR: $X_t = \left(1-B_t \right) \circ M + B_t \circ X_T$, where $M$ is a masked image, $B_t^{i}, i=1,…,d$ are i.i.d Bernouli$\left(\frac{t}{T}\right)$ random variables, and $\circ$ is the Hadamard product.
>
> Parametization:
> * DTM: $Y=X_T-X_0$.
> * MAR: $Y=X_T$.
>
> Hence, (i) the DTM model’s input ($X_t$) is a noisy image while the MAR model’s input ($X_t$) is an image where some patches are clean and some are masked. (ii) DTM model’s output ($Y$) is a direction $X_T-X_0$ intersecting $X_t$ while MAR model’s output ($Y$) is an estimation of the clean image. (iii) On generation, DTM is **non-AR** and predicts a direction and takes a small step in that direction, changing **all patches at each step**. In contrast, MAR is **arbitrary-order-AR** that predicts a clean image and masks back some of the patches, thus changing only a **“set of tokens” at each step**. A set of tokens of size one leads to an autoregressive model in the order of patch generation.
>
>
> > While the authors propose modeling the latent difference, rather than the conventional $\eps$-prediction in diffusion models or velocity prediction in flow models, I am not convinced why this is needed and such a design choice should give better results.
>
> First we would like to emphasize that in practice, as shown in the paper, DTM is **considerably favorable** to the flow matching (FM) method.
>
> Second, as the reviewer is probably aware, all diffusion-type methods reproduce the training set at a global minimum loss, therefore the hope of theoretically distinguishing generalization abilities among them is probably futile. However, we can offer an intuition: Given a current state $X_t=x_t$, FM learns to approximate the **expected transition** $Y_t = \mathbb{E}[X_T-X_0|X_t=x_t]$ while DTM learns to **sample the underlying distributions of transitions** $Y_t \sim p_{Y|t}(\cdot|X_t=x_t)$. We hypothesize that in the large scale settings (as in our case) where model capacity is not a constraint, the more elaborate supervision of DTM is beneficial for the model training.
>
> > Similarly, the use of independent linear process is not justified. What does it exactly mean by a better regularity of the conditional reverse kernel?
>
> We respectfully disagree that the independent process is unjustified.
>
> First we refer the reviewer to Figure 10 in the Appendix that shows a significant advantage for the independent linear process, therefore using this process is **justified empirically**.
>
> Second, regarding regularity, given a current state $X_t=x_t$, Figure 6 in the paper illustrates that the support of the transition kernel of the independent linear process (equation 15) is significantly larger than the support of the linear process (equation 10). In practice, using the linear process with TM steps$>1$ for training an autoregressive kernel results in trivial overfitting (since a linear interpolant is fully determined by any two points on the trajectory) and leads to poor results. Thus, the use of the independent linear process which has a wider support results in a better regularized training objective and leads to state-of-the-art results.
>
> > The authors show that DTM achieves the best performances to full sequence denoising such as FM. However, I find the results contradictory since the DTM also outperforms AR-based methods that adopt causal attention.
>
> We are not sure we understand the reviewer’s comment, why “DTM outperforms AR-based” methods is contradictory? As we found in our experiments, which were performed in a controlled and fixed settings, and is also shown in [5] that the reviewer mentions, we found AR (causal) methods to be sub-par in general to non-AR (bidirectional) methods in generated image quality.
>
>
> > Additionally, how does DTM compete against FM in inference time or training stability?
>
> To show inference time of DTM versus FM we have conducted several experiments and added several relevant Tables **in Reviewer’s ag7i response**.
>
> Overall, Table 1 in Reviewer’s ag7i response compares the wall-clock time of FM and DTM, and in particular DTM (16 backbone NFE and 4 head NFE) can **achieve superior performance** to FM (128 backbone NFE) with a **7x speedup**.
>
> In more detail, Tables 2(a–b) in Reviewer’s ag7i response present the dependence of DTM’s CLIPScore and PickScore on the number of function evaluations (NFE) in the flow head and backbone. Table 4 reports the corresponding inference times. Tables 3(a–b) in Reviewer’s ag7i response present flow matching’s (FM) CLIPScore and PickScore as a function of backbone NFE.
>
> Regarding training stability, the reported DTM and FM model are trained with the exact same hyper-parameters and training steps, and we didn’t observe any stability issues.
>
> > I could not find limitations nor societal impact of the work in the paper.
>
> The limitations and societal impact statement appears in the conclusions section in lines (286-287) and lines (289-290). We will modify/elaborate the limitations to include “The improved performance of ARTM/FHTM comes at the price of a higher sampling cost, i.e., NFE counts are proportional to the number of transition steps, see e.g., Table 1 in the paper.”

---

> > ### Author Response · Authors · 2025-08-06
> >
> > As discussion period coming to an end - we would like to draw the reviewer's attention to the comprehensive rebuttal and new results we produced to address their comments and would greatly appreciate a response. Thanks!

---

> > > ### Comment · Reviewer_kPcj · 2025-08-06
> > >
> > > Appreciate the authors for resolving most of the concerns.
> > >
> > > Here are further clarification of my previous comments.
> > > - From my understanding, DTM and FM are the same in a sense they both use bi-direction attention and condition on the previous sample $x_{t}$ to sample $x_{t+1}$. Then is the performance boost (FM to DTM) coming from the architectural advantages or parameterization? What is component leading to the major performance improvements?
> > > - Would FM perform better if we used independent linear for the supervising process?
> > > - Timestep $T$ and $1$ are used interchangeably. This confused the notations whether $x_1$ indicating one-step or fully generated sample (L125).
> > > - With some typos in the algorithms (Fig. 20-22): rand_like -> randn_like.
> > > - Some discrepancies between the figures and algorithms. The velocity head in Alg. 7 FHTM seems to be both conditioned on $t, s$ (subscript), but the notation in Fig. 5 implies it is conditioned only on $s$.
> > > - From Fig. 5, it was not clear to me how the initial tokens were sampled ($x_0$) from boi. Does it condition on previous Gaussian patches to sample Gaussian samples? Sampling Gaussian from a velocity head seemed unintuitive to me.
> > >
> > > These are the some of the points I found hard to understand the method. I would appreciate if the authors can provide some clarifications.

---

> > > > ### Author Response · Authors · 2025-08-06
> > > >
> > > > We are pleased to hear that most of the reviewer's concerns have been addressed. Below, we provide further clarifications.
> > > >
> > > > > From my understanding, DTM and FM are the same in a sense they both use bi-direction attention and condition on the previous sample $x_t$ to sample $x_{t+1}$ . Then is the performance boost (FM to DTM) coming from the architectural advantages or parameterization? What is component leading to the major performance improvements?
> > > >
> > > > We would like to emphasize that DTM and FM both use the same linear supervising process $X_t=(1-\frac{t}{T})X_0 + \frac{t}{T}X_T$ and the same parametrization $Y=X_T-X_0$, however they are distinct in their modeling. Given a current state $X_t=x_t$:
> > > >
> > > > 1. **FM**: at each step FM takes as input the current state $X_t=x_t$ and output the velocity which is exactly equal to the expectation of $Y=X_T-X_0$ conditioned on $X_t=(1-\frac{t}{T})X_0 + \frac{t}{T}X_T=x_t$ (denoted before as $\mathbb{E}[Y|X_t=x_t]$).
> > > >
> > > > 2. **DTM**: at each step DTM takes as input a current state $X_t=x_t$ and an output a sample of $Y=X_T-X_0$ conditioned on $X_t=(1-\frac{t}{T})X_0 + \frac{t}{T}X_T=x_t$.
> > > >
> > > > To learn a sampler of $Y$ conditioned on $X_t=x_t$ in a scalable manner DTM must use a flow head. The flow head is an additional small generative model parametrized by a 40M parameters MLP and trained (end-to-end with the backbone) with flow matching loss where the target is the distribution of $Y$ conditioned on $X_t=x_t$ as in Algorithm 3. in the paper.
> > > >
> > > > To summarize, we believe that the addition of the flow head which amounts to architectural and loss change compared to FM is the reason for improved performance.
> > > >
> > > > > Would FM perform better if we used independent linear for the supervising process?
> > > >
> > > > The independent linear process **cannot be used as supervision for FM**. The intuition can be drawn from Figure 6 in the paper. Given a current state $X_t=x_t$ (since $X_{0,t+1}$ is independent of $X_t$), the line connecting $X_{0,t+1}$ and $X_T$ does not necessarily cross  $X_t$. Thus in the limit of continuous time, the transition kernel of the independent linear process will involve a jump probability and cannot be modeled solely by a velocity field (as done in FM).
> > > >
> > > > > Timestep $T$ and $1$  are used interchangeably. This confused the notations whether $x_1$  indicating one-step or fully generated sample (L125).
> > > >
> > > > We apologize for the confusion in line 125, indeed it should note $s=1$ instead of $t=1$ indicating a sample from $B_1$, which in the case of DTM would be $B_1=Y$.  Will fix this in the camera ready version.
> > > >
> > > > > With some typos in the algorithms (Fig. 20-22): rand_like -> randn_like.
> > > >
> > > > Thank you for noting this, we will fix it in the camera ready version.
> > > >
> > > > > Some discrepancies between the figures and algorithms. The velocity head in Alg. 7 FHTM seems to be both conditioned on $t,s$ (subscript), but the notation in Fig. 5 implies it is conditioned only on $s$.
> > > >
> > > > Thanks for raising this typo, indeed Figure 5 is the correct one (as is in equation 18). We will remove the $t$ subscript for the head in Alg. 7 in the camera ready version. We also note that $t$ is implicitly given through the hidden state $h_t$ and in practice further experiments show that explicitly input $t$ into the flow head is redundant.
> > > >
> > > > > From Fig. 5, it was not clear to me how the initial tokens were sampled ($x_0$) from boi. Does it condition on previous Gaussian patches to sample Gaussian samples? Sampling Gaussian from a velocity head seemed unintuitive to me.
> > > >
> > > > The initial token $X_0$ for **DTM** and **ARTM** is sampled from a Gaussian noise, i.e., $X_0\sim \mathcal{N}(0,I)$. While for FHTM $X_0$ is constant and always taken to be a “boi” token.
> > > >
> > > > We hope this answers your questions and we would be happy to clarify any further concerns.

---

> > > > > ### Comment · Reviewer_kPcj · 2025-08-06
> > > > >
> > > > > Thank you to authors for addressing the comments. But it is confusing what the authors mean by "While for FHTM $X_0$ is constant and always taken to be a “boi” token." How can $X_0$ be a constant? I believed this should also be sampled from random Gaussian. Could the authors clarify how exactly the sampling is done for FHTM?

---

> > > > > > ### Author Response · Authors · 2025-08-06
> > > > > >
> > > > > > > Thank you to authors for addressing the comments. But it is confusing what the authors mean by "While for FHTM $X_0$ is constant and always taken to be a “boi” token." How can $X_0$ be a constant? I believed this should also be sampled from random Gaussian. Could the authors clarify how exactly the sampling is done for FHTM?
> > > > > >
> > > > > > We are sorry for the confusion, let us explain.
> > > > > > When training FHTM (Figure 5) the head and loss is taken only for $t \geq 1$. That is,  $h_0^1,\ldots,h_0^n$ do not have gradients. This is also consistent with equation 17.
> > > > > >
> > > > > > Sampling from FHTM is done by concatenating [“boi”,$X_0$] and then sampling consecutively $X_1^1, X_1^2,\ldots,X_1^n,X_2^1,\ldots,X_2^n,X_3^1,\ldots$ one after the other with causal attention (as standard in AR).
> > > > > >
> > > > > > In practice, concatenating $X_0$ serves as dummy tokens since, as described above, no loss is taken on $h^1_0,\ldots,h_0^n$.  Therefore, it is sufficient (and more efficient) to take $X_0$ as “boi” (i.e., “constant”), see pseudocode in Figure 22. We will also clarify this in the paper.
> > > > > >
> > > > > > We would be happy to clarify further if the above is still not clear and/or try to address any remaining concerns of the reviewer.

---

> > > > > > > ### Comment · Reviewer_kPcj · 2025-08-06
> > > > > > >
> > > > > > > Thank you for the clarification. That makes much more sense. I believe the general framework presented will have great impact in generative modeling community. I'm now convinced that this paper should be presented at the conference. I will revise my score accordingly. Thank you to the authors for providing clarifications.

---

> > > > > > > > ### Author Response · Authors · 2025-08-07
> > > > > > > >
> > > > > > > > We thank the reviewer for their engagement and positive feedback and happy to learn that they are willing to update their score accordingly.

---

### Official Review · Reviewer_jNpW · 2025-07-03

**Clarity:** 3
**Significance:** 3
**Originality:** 3
**Rating:** 5
**Confidence:** 4

**Summary:**

The paper proposes a novel auto-regressive generative model based on "Transition Matching". Here, the idea is to form a Markov model that takes some easily sampled data to the target distribution gradually based on a fixed process called "supervising process". The idea is then to use various transition modeling strategies and train them to generate data at the test time. This results in a generic and rich set of generative models, for which diffusion models and flow matching could be considered as its special cases. Because of the generality of the model, one would expect a greater flexibility (e.g. could be made auto-regressive as a result of the modeling choices) and potentially improved generative capabilities compared to previous models whose design choices are saturated for the most part. Three variants of the proposed model was trained on massive datasets "350M licensed Shutterstock" and tested based on PartiPrompts and MS-COCO benchmarks, based on various metrics such as CLIPScore, PickScore, etc.

**Questions:**

Please see above.

**Ethical Concerns:**

["NO or VERY MINOR ethics concerns only"]

**Final Justification:**

All my concerns are addressed.

**Quality:**

3

**Strengths And Weaknesses:**

The paper overall looks promising and clear. I have few concerns:

- The problem of text-to-image generation is multi-facet problem and may previous work focused on improving specific aspects, e.g. compositionality, quality, bias, style transfer, etc. It is not clear which of these aspects is expected to be improved by the proposed method. With regard to compositionaliity, one would expect benchmarks such as T2I-CompBench and other related ones to be tested against. Furthermore, many of the proposed metrics lack alignment with human perception, making results less reliable.
- The fact that DTM (Difference Transition Matching) gave better results compared to more complex ARTM and FHTM sounds puzzling and was not discussed in details in the paper. My question to the authors is why DTM outperforms such models.
- As the authors mentioned in line 286 "The improved performance of DTM/ARTM/FHTM comes at the price of a higher sampling cost". But we know certain test-time scaling methods, such as noise optimization, that improve the image generation quality by large margins when given higher test-time compute budget (e.g. see ReNO). The authors should elaborate how their method compare to such schemes.

---

> ### Author Rebuttal · Authors · 2025-07-30
>
> > With regard to compositionality, one would expect benchmarks such as T2I-CompBench and other related ones to be tested against.
>
> We have followed the reviewer’s suggestion on the benchmark they recommended and evaluated our models and baseline on the T2I-CompBench. Results are provided below in Table 7 and the global ranking of each method (defined as the sum of its ranking across all benchmarks: GenEval, T2I-Compbench, PartiPrompts, and MS-COCO) is shown in Table 8 Note that the new benchmark does not change the fact that DTM is best among non-AR and FHTM is best among AR methods.
>
> For tables 7-8 below: Best score is denoted by **Bold** with ★ and second best is denoted by **Bold** only.  NFE\* is the number of function evaluations in the backbone. † Indicates inference with activation caching.
>
> **Table 7.** T2I-CompBench
> |Kernel|Color|Shape|Texture|2D-Spatial|3D-Spatial|Numeracy|Non-Spatial|Complex|
> |---|---|---|---|---|---|---|---|---|
> |MAR-discrete|0.6666|0.4535|0.5316|0.1474|0.2693|0.4538|**0.3090**|0.3096|
> |MAR|**0.7378★**|**0.5174★**|**0.6588**|0.1638|0.3002|0.4962|0.3082|**0.3392★**|
> |MAR-Fluid|0.6997|0.4768|0.6149|0.1454|0.2938|0.4681|0.3037|0.3289|
> |FM|0.6855|0.4511|0.5615|0.1372|0.2706|0.4526|0.3026|0.3138|
> |DTM|**0.7316**|**0.4865**|**0.6597★**|**0.1839★**|**0.3113★**|**0.5043★**|0.3075|**0.3382**|
> |AR-discrete|0.6068|0.4757|0.5958|0.1095|0.2535|0.4423|**0.3098★**|0.3097|
> |AR|0.5062|0.3669|0.5061|0.1041|0.2441|0.4210|0.2989|0.2983|
> |ARTM-2|0.6520|0.4430|0.5870|0.1475|0.2748|0.4800|0.3074|0.3267|
> |ARTM-3|0.6555|0.4738|0.5842|0.1459|0.2832|0.4855|0.3062|0.3227|
> |FHTM-2|0.6318|0.4318|0.5730|0.1403|0.2818|0.4830|0.3058|0.3229|
> |FHTM-3|0.6604|0.4640|0.5839|0.1394|0.2755|0.4810|0.3066|0.3223|
> |FHTM-3LLM|0.6166|0.4618|0.5945|**0.1688**|**0.3081**|**0.5010**|0.3079|0.3310|
>
> **Table 8.** Global ranking: For metric in GenEval, T2I-Compbench, PartiPrompts, and MS-COCO  we rank all models from 1 to 12 (where 1 is best model) and we sum the ranks from all metrics. Lower is better.
> |Kernel|Rank↓|
> |---|---|
> |MAR-discrete|200|
> |MAR|127|
> |MAR-Fluid|220|
> |FM|179|
> |DTM|**58★**|
> |AR-discrete|245|
> |AR|321|
> |ARTM-2|184|
> |FHTM-2|185|
> |ARTM-3|130|
> |FHTM-3|130|
> |FHTM-3 LLM|**99**|
>
>
> > The problem of text-to-image generation is multi-facet problem and many previous work focused on improving specific aspects, e.g. compositionality, quality, bias, style transfer, etc. It is not clear which of these aspects is expected to be improved by the proposed method ;  … Furthermore, many of the proposed metrics lack alignment with human perception, making results less reliable.
>
> We have collected standard and widely used metrics found in similar papers including for example:
>
> GenEval eval in: SD3 [1], Fluid [2], Transfusion [3], EMU3 [4], Janus [5], Flow-GRPO [6].
>
> CLIPScore eval in: SD3 [1], Transfusion [3], EMU3 [4],  DALLE-3 [7], DPO [8], SDXL [9], Imagen [10]
>
> PIckScore eval in: Flow-GRPO [6], DPO [8].
>
> We now added T2I compBench.
> If the reviewer feels we have missed other well accepted, open-source and open-license auto-evals we would be happy to incorporate them as-well.
>
>
> > The fact that DTM (Difference Transition Matching) gave better results compared to more complex ARTM and FHTM sounds puzzling and was not discussed in detail in the paper. My question to the authors is why DTM outperforms such models.
>
> First, we would not characterize DTM as “simpler” or “less performant” than ARTM/FHTM. If anything, its bidirectional attention makes it more expressive and in our empirical  apples-to-apples comparison it was favorable.
>
>
> > As the authors mentioned in line 286 "The improved performance of DTM/ARTM/FHTM comes at the price of a higher sampling cost". But we know certain test-time scaling methods, such as noise optimization, that improve the image generation quality by large margins when given higher test-time compute budget (e.g. see ReNO). The authors should elaborate how their method compares to such schemes.
>
> **Regarding time-performance of DTM:**
> DTM is in fact **considerably faster** than FM. We have conducted several experiments and added several relevant Tables **in Reviewer’s ag7i response**.
> Mainly, Table 1 in Reviewer’s ag7i response compares the wall-clock time of FM and DTM, and in particular DTM (16 backbone NFE and 4 head NFE) can **achieve superior performance** to FM (128 backbone NFE) with a **7x speedup**.
> In more detail, Tables 2(a–b) in Reviewer’s ag7i response present the dependence of DTM’s CLIPScore and PickScore on the number of function evaluations (NFE) in the flow head and backbone. Table 4 in Reviewer’s ag7i response reports the corresponding inference times. Tables 3(a–b) present flow matching’s (FM) CLIPScore and PickScore as a function of backbone NFE.
>
>
> **Regarding test-time scaling methods:**
> Nevertheless, we did compare to one rather popular test-time scaling method Restart [11], see Tables 5.(a-b) and 6 **in Reviewer’s QC47 response**. Note that although Restart (with up to 2400 NFE) shows some improvement over FM it is still *considerably sub-par* to DTM.
> ReNO [12] utilizes an additional reward loss and therefore cannot be considered a strictly apples-to-apples comparison. Incorporating reward based optimization to TM is, however, a very interesting future research avenue!
>
> **Regarding ARTM/FHTM test-time scaling:**
> We are not aware of any useful test-time scaling methods for AR image generation. The ARTM/FHTM variants do not offer any improvement in sampling efficiency, however compared to the AR baseline they achieve far superior performance. And we do not claim these variants outperform the FM baseline.
>
> [1] Esser, Patrick, et al. "Scaling rectified flow transformers for high-resolution image synthesis." Forty-first international conference on machine learning. 2024.
>
> [2] Fan, Lijie, et al. "Fluid: Scaling autoregressive text-to-image generative models with continuous tokens." arXiv preprint arXiv:2410.13863 (2024).
>
> [3] Zhou, Chunting, et al. "Transfusion: Predict the next token and diffuse images with one multi-modal model." arXiv preprint arXiv:2408.11039 (2024).
>
> [4] Wang, Xinlong, et al. "Emu3: Next-token prediction is all you need." arXiv preprint arXiv:2409.18869 (2024).
>
> [5] Chen, Xiaokang, et al. "Janus-pro: Unified multimodal understanding and generation with data and model scaling." arXiv preprint arXiv:2501.17811 (2025).
>
> [6] Liu, Jie, et al. "Flow-grpo: Training flow matching models via online rl." arXiv preprint arXiv:2505.05470 (2025).
>
> [7] Betker, James, et al. "Improving image generation with better captions." Computer Science. openai. com/papers/dall-e-3. pdf 2.3 (2023):8.
>
> [8] Wallace, Bram, et al. "Diffusion model alignment using direct preference optimization." Proceedings of the IEEE/CVF Conference on Computer Vision and Pattern Recognition. 2024.
>
> [9] Podell, Dustin, et al. "Sdxl: Improving latent diffusion models for high-resolution image synthesis." arXiv preprint arXiv:2307.01952 (2023).
>
> [10] Saharia, Chitwan, et al. "Photorealistic text-to-image diffusion models with deep language understanding." Advances in neural information processing systems 35 (2022): 36479-36494.
>
> [11] Xu, Yilun, et al. "Restart sampling for improving generative processes." Advances in Neural Information Processing Systems 36 (2023): 76806-76838.
>
> [12] Eyring, Luca, et al. "Reno: Enhancing one-step text-to-image models through reward-based noise optimization." Advances in Neural Information Processing Systems 37 (2024): 125487-125519.

---

> ### Comment · Reviewer_jNpW · 2025-08-06
>
> Thanks for addressing my major concerns. I would be happy to raise my score.

---

### Official Review · Reviewer_QC47 · 2025-07-05

**Clarity:** 3
**Significance:** 3
**Originality:** 3
**Rating:** 5
**Confidence:** 3

**Summary:**

This paper presents Transition Matching (TM), a new class of generative models that merges the strengths of diffusion and flow-based approaches. TM broadens the generative design space by introducing non-deterministic probabilistic transitions and flexible, non-consecutive supervision. The authors develop three variants: Difference (DTM), Autoregressive (ARTM), and Full History (FHTM) Transition Matching. DTM enhances image quality and text consistency while speeding up sampling. As partially and fully causal models, ARTM and FHTM achieve generation quality on par with or exceeding non-causal methods and integrate easily with existing text generation technologies. In particular, FHTM is the first fully causal model in the continuous domain to outperform flow-based methods, excelling in text-to-image synthesis. The paper validates the advantages of the TM framework through extensive comparisons and outlines future work on improving efficiency and integrating FHTM into larger multimodal systems.

**Questions:**

1. Has this method been validated on large-scale generative models?

**Ethical Concerns:**

["NO or VERY MINOR ethics concerns only"]

**Final Justification:**

Thank you for your response. The authors have addressed some of my concerns, and I appreciate the thorough exploratory experiments conducted in the paper. I have decided to raise my score to 5.

**Limitations:**

yes.

**Paper Formatting Concerns:**

Paper Checklist is not fully matched the requirements.

**Quality:**

3

**Strengths And Weaknesses:**

### **Strengths:**

1. The paper presents a novel generative model called Transition Matching (TM) that combines the advantages of diffusion and flow models. The authors introduce non-deterministic probability transfer kernels and arbitrary non-continuous supervised processes, significantly expanding the design space. They propose three different TM variants - Difference Transition Matching (DTM), Autoregressive Transition Matching (ARTM), and Full History Transition Matching (FHTM) - each achieving performance optimization in specific aspects.

2. The proposed TM framework offers a powerful tool for generating high-quality samples while maintaining computational efficiency. It addresses limitations of existing approaches and provides a new direction for future research in the field of generative modeling. The authors highlight the significance of their work in terms of its potential impact on various applications such as image generation, text-to-image synthesis, and multi-modal systems.

### **Weaknesses:**

1. As authors stated, the improved performance of DTM/ARTM/FHTM comes at the price of a higher sampling cost.

2. The analysis of the model's performance is simplistic.

---

> ### Author Rebuttal · Authors · 2025-07-30
>
> > As authors stated, the improved performance of DTM/ARTM/FHTM comes at the price of a higher sampling cost.
>
> DTM is in fact **considerably faster** than FM. We have conducted several experiments and added several relevant Tables in **Reviewer’s ag7i response**.
> Mainly, Table 1 in Reviewer’s ag7i response compares the wall-clock time of FM and DTM, and in particular DTM (16 backbone NFE and 4 head NFE) can **achieve superior performance** to FM (128 backbone NFE) with a **7x speedup**.
> In more detail, Tables 2(a–b) in Reviewer’s ag7i response present the dependence of DTM’s CLIPScore and PickScore on the number of function evaluations (NFE) in the flow head and backbone. Table 4 Reviewer’s ag7i response reports the corresponding inference times. Tables 3(a–b) in Reviewer’s ag7i response present flow matching’s (FM) CLIPScore and PickScore as a function of backbone NFE.
>
>
> > The analysis of the model's performance is simplistic.
>
> We kindly ask the reviewer to be more specific. We have made an honest effort to collect all popular benchmarks and auto-metric that are open-source and open-license: The GenEval benchmark and additional 6 auto-metrics: CLIPScore, PickScore, ImageReward, UnifiedReward, Aesthetic, and DeQA Score, where the metrics are evaluated on two benchmarks: MS-COCO and PartiPrompts.
> The GenEval benchmark is widely accepted metric for evaluating text-to-image and has been used by many acclaimed previous works: SD3 [1], Fluid [2], Transfusion [3], EMU3 [4], Janus [5], Flow-GRPO [6], and so does the metrics we report. For example, CLIPScore has been used by SD3 [1], Transfusion [3], EMU3 [4],  DALLE-3 [7], DPO [8], SDXL [9], Imagen [10], and PickScore by Flow-GRPO [6], DPO [8].
>
> We also added during the rebuttal the benchmark suggested by Reviewer jNpW: T2I-CompBench. See Table 7  in **Reviewer’s jNpW response**.
>
> If the reviewer feels we have missed other well known, open-source and open-license auto-evals we would be happy to incorporate them as-well.
>
> Lastly, we also added two fully discrete baselines: discrete-AR and discrete-MAR using the tokenizer of Chameleon [11]. The results are summarized below in Tables 5.(a-b) and 6, and are consistent with the submitted paper’s claims. To summarize all the evaluations we add Table 8 **in Reviewer’s jNpW response** showing the global ranking of each method (defined as the sum of its ranking across all benchmarks: GenEval, T2I-Compbench, PartiPrompts, and MS-COCO ).
>
> For tables 5-6 below: Best score is denoted by **Bold** with ★ and second best is denoted by **Bold** only.  NFE\* is the number of function evaluations in the backbone. † Indicates inference with activation caching.
>
> **Table 5.a**: Main Table. PartiPrompts
> ||Attention|Kernel|Arch|NFE*|CLIPScore↑|PickScore↑|ImageReward↑|UnifiedReward↑|Aesthetic↑|DeQAScore↑|
> |---|---|---|---|---|---|---|---|---|---|---|
> |Baseline|Full|MAR-discrete|DiT|256|26.80|20.70|0.14|4.31|5.15|2.48|
> |Baseline|Full|MAR|DiT|256|**27.00★**|20.70|0.33|4.26|4.95|2.36|
> |Baseline|Full|MAR-Fluid|DiT|256|26.00|20.50|0.07|3.82|4.74|2.36|
> |Baseline|Full|FM|DiT|256|26.00|21.00|0.23|4.78|5.29|**2.55**|
> |Baseline|Full|FM|DiT|2400|26.00|**21.10**|0.24|4.81|5.29|**2.55**|
> |Baseline|Full|FM-Restart|DiT|2400|26.10|**21.10**|0.34|4.83|**5.31**|2.53|
> |TM|Full|DTM|DiT|32|26.80|**21.20★**|**0.53★**|**5.12★**|**5.42★**|**2.65★**|
> |Baseline|Causal|AR-discrete†|DiT|256|26.70|20.40|-0.01|3.74|4.81|2.38|
> |Baseline|Causal|AR†|DiT|256|24.90|20.10|-0.43|3.41|4.50|2.27|
> |TM|Causal|ARTM–2†|DiT|2×256|26.80|20.80|0.29|4.49|5.03|2.37|
> |TM|Causal|FHTM–2†|DiT|2×256|26.80|20.80|0.30|4.59|5.13|2.44|
> |TM|Causal|ARTM–3†|DiT|3×256|**27.00**|20.90|0.38|4.77|5.21|2.53|
> |TM|Causal|FHTM–3†|DiT|3×256|**27.00**|20.90|0.31|4.77|5.15|2.44|
> |TM|Causal|FHTM–3†|LLM|3×256|**27.00**|21.00|**0.43**|**5.02**|5.30|2.54|
>
>
> **Table 5.b**: Main Table. MS-COCO
> |Model|Attention|Kernel|Arch|NFE*|CLIPScore↑|PickScore↑|ImageReward↑|UnifiedReward↑|Aesthetic↑|DeQAScore↑|
> |---|---|---|---|---|---|---|---|---|---|---|
> |Baseline|Full|MAR-discrete|DiT|256|**26.57**|20.63|0.01|4.14|5.27|2.41|
> |Baseline|Full|MAR|DiT|256|26.12|20.66|0.17|4.62|5.06|2.34|
> |Baseline|Full|MAR-Fulid|DiT|256|25.46|20.45|-0.11|3.94|4.86|2.38|
> |Baseline|Full|FM|DiT|256|25.78|**21.11**|0.09|5.00|5.45|2.47|
> |Baseline|Full|FM|DiT|2400|25.78|**21.11**|0.09|5.00|5.45|2.47|
> |Baseline|Full|FM-Restart|DiT|2400|25.78|**21.11**|0.15|5.11|5.48|2.44|
> |TM|Full|DTM|DiT|32|26.16|**21.19★**|**0.22**|**5.38**|**5.55★**|**2.58★**|
> |Baseline|Causal|AR-discrete†|DiT|256|**26.69★**|20.31|-0.06|3.83|4.93|2.34|
> |Baseline|Causal|AR†|DiT|256|24.83|20.11|-0.48|3.60|4.76|2.34|
> |TM|Causal|ARTM-2†|DiT|2×256|25.90|20.75|0.07|4.70|5.19|2.41|
> |TM|Causal|FHTM-2†|DiT|2×256|25.91|20.79|0.07|4.78|5.27|2.44|
> |TM|Causal|ARTM-3†|DiT|3×256|26.07|20.92|0.11|4.99|5.35|2.46|
> |TM|Causal|FHTM-3†|DiT|3×256|26.14|20.98|0.15|5.23|5.38|2.41|
> |TM|Causal|FHTM-3†|LLM|3×256|26.14|21.08|**0.24★**|**5.51★**|**5.53**|**2.51**|
>
>
> **Table 6.**: Main Table. GenEval
> ||Attention|Kernel|Arch|NFE*|Overall↑|Single-object↑|Two-objects↑|Counting↑|Colors↑|Position↑|ColorAttribute↑|
> |---|---|---|---|---|---|---|---|---|---|---|---|
> |Baseline|Full|MAR-discrete|DiT|256|0.44|0.86|0.43|0.37|0.66|0.13|0.29|
> |Baseline|Full|MAR|DiT|256|**0.52**|**0.98★**|0.56|**0.43**|0.73|0.11|**0.38**|
> |Baseline|Full|MAR-Fluid|DiT|256|0.44|0.90|0.33|0.37|0.76|0.12|0.28|
> |Baseline|Full|FM|DiT|256|0.47|0.91|0.52|0.27|0.71|0.12|0.34|
> |Baseline|Full|FM|DiT|2400|0.47|0.91|0.51|0.25|0.72|0.14|0.36|
> |Baseline|Full|FM-Restart|DiT|2400|0.49|0.89|**0.59★**|0.29|0.73|0.13|**0.38**|
> |TM|Full|DTM|DiT|32|**0.54★**|0.93|**0.58**|0.35|**0.79★**|**0.20★**|**0.46★**|
> |Baseline|Causal|AR-discrete†|DiT|256|0.41|**0.96**|0.40|0.33|0.60|0.07|0.19|
> |Baseline|Causal|AR†|DiT|256|0.34|0.86|0.26|0.15|0.63|0.06|0.15|
> |TM|Causal|ARTM-2†|DiT|2×256|0.49|0.95|0.51|0.39|**0.79★**|0.11|0.27|
> |TM|Causal|FHTM-2†|DiT|2×256|0.48|**0.96**|0.48|0.25|**0.78**|0.09|0.37|
> |TM|Causal|ARTM-3†|DiT|3×256|0.51|0.95|0.54|0.41|**0.79★**|0.16|0.28|
> |TM|Causal|FHTM-3†|DiT|3×256|**0.52**|**0.98★**|0.54|**0.44★**|0.74|0.16|0.34|
> |TM|Causal|FHTM-3†|LLM|3×256|0.49|0.94|0.55|0.37|0.69|**0.17**|0.29|
>
>
> > Has this method been validated on large-scale generative models?
>
> We would like to bring to the reviewer’s attention that the results reported in the paper include 10 models of 1.7B parameters, trained from scratch on a large dataset of 350M image-caption pairs for 500k training iterations. We believe this is **among the largest, most comprehensive and fair large-scale evaluation openly available**. For comparison, SD3 [1], which we consider the largest available fair comparison of diffusion and flow models, made most of their evaluation efforts on models with 2B parameters or less, and trained only a single model with 8B parameters.
>
> [1] Esser, Patrick, et al. "Scaling rectified flow transformers for high-resolution image synthesis." Forty-first international conference on machine learning. 2024.
>
> [2] Fan, Lijie, et al. "Fluid: Scaling autoregressive text-to-image generative models with continuous tokens." arXiv preprint arXiv:2410.13863 (2024).
>
> [3] Zhou, Chunting, et al. "Transfusion: Predict the next token and diffuse images with one multi-modal model." arXiv preprint arXiv:2408.11039 (2024).
>
> [4] Wang, Xinlong, et al. "Emu3: Next-token prediction is all you need." arXiv preprint arXiv:2409.18869 (2024).
>
> [5] Chen, Xiaokang, et al. "Janus-pro: Unified multimodal understanding and generation with data and model scaling." arXiv preprint arXiv:2501.17811 (2025).
>
> [6] Liu, Jie, et al. "Flow-grpo: Training flow matching models via online rl." arXiv preprint arXiv:2505.05470 (2025).
>
> [7] Betker, James, et al. "Improving image generation with better captions." Computer Science. openai. com/papers/dall-e-3. pdf 2.3 (2023):8.
>
> [8] Wallace, Bram, et al. "Diffusion model alignment using direct preference optimization." Proceedings of the IEEE/CVF Conference on Computer Vision and Pattern Recognition. 2024.
>
> [9] Podell, Dustin, et al. "Sdxl: Improving latent diffusion models for high-resolution image synthesis." arXiv preprint arXiv:2307.01952 (2023).
>
> [10] Saharia, Chitwan, et al. "Photorealistic text-to-image diffusion models with deep language understanding." Advances in neural information processing systems 35 (2022): 36479-36494.
>
> [11]  Team, Chameleon. "Chameleon: Mixed-modal early-fusion foundation models." arXiv preprint arXiv:2405.09818 (2024).

---

> > ### Author Response · Authors · 2025-08-06
> >
> > As discussion period coming to an end - we would like to draw the reviewer's attention to the comprehensive rebuttal and new results we produced to address their comments and would greatly appreciate a response. Thanks!

---

### Official Review · Reviewer_ag7i · 2025-07-08

**Clarity:** 3
**Significance:** 3
**Originality:** 3
**Rating:** 4
**Confidence:** 1

**Summary:**

This paper introduces Transition Matching (TM), a general framework for generating discrete-time continuous-state sequences. TM unifies both flow-based methods and continuous autoregressive generation. The framework explores three novel variants: Difference Transition Matching, Autoregressive Transition Matching, and Full History Transition Matching. Experiments demonstrate that FHTM outperforms flow-based methods in the text-to-image task for continuous domains.

**Questions:**

Please refer to the weaknesses.

**Ethical Concerns:**

["NO or VERY MINOR ethics concerns only"]

**Limitations:**

I’m unable to find the discussion about the limitation except line 157.

**Quality:**

3

**Strengths And Weaknesses:**

**Strengths**

1. ARTM and FHTM generate continuous causal AR with quality comparable to non-causal approaches.

2. The proposed methods achieve state-of-the-art image quality and text adherence in text-to-image generation. They significantly improve text adherence compared to Flow matching and set a new state-of-the-art on this task.

**Weaknesses**

Experiments indicate that the Non-Formal Error (NFE) for ARTM/FHTM is high. Further analysis or experiments specifically focusing on the trade-off between quality and NFE for these causal models would be beneficial.

---

> ### Author Rebuttal · Authors · 2025-07-30
>
> > Experiments indicate that the Non-Formal Error (NFE) for ARTM/FHTM is high. Further analysis or experiments specifically focusing on the trade-off between quality and NFE for these causal models would be beneficial.
>
> We assume the reviewer means Number of Function Evaluations (NFE).
> DTM is in fact **considerably faster** than FM. We have conducted several experiments and added several relevant Tables below.
>
> Mainly, Table 1 compares the wall-clock time of FM and DTM, and in particular DTM (16 backbone NFE and 4 head NFE) can **achieve superior performance** to FM (128 backbone NFE) with a **7x speedup**.
> In more detail, Tables 2(a–b) present the dependence of DTM’s CLIPScore and PickScore on the number of function evaluations (NFE) in the flow head and backbone. Table 4 reports the corresponding inference times. Tables 3(a–b) present flow matching’s (FM) CLIPScore and PickScore as a function of backbone NFE.
>
>
> **Table 1.**: DTM vs FM sampling time.
> | Kernel | Time (sec) | CLIPScore | PickScore |
> | ------ | ---------- | --------- | --------- |
> | FM     | 10.8       | 26.0      | 21.0      |
> | DTM    | **1.6**    | **26.8**  | **21.1**  |
>
>
>
> **Table 2.a**: DTM CLIPScore,  Head NFE (Rows) vs. TM steps (Columns).
> |HeadNFE/TMsteps|1|2|4|8|16|32|64|128|
> |--------|----|----|----|----|----|----|----|----|
> |1|15.8|17.0|20.4|22.8|23.2|23.2|23.0|22.8|
> |2|16.1|18.6|24.2|26.2|26.4|26.4|26.2|26.2|
> |4|17.9|21.1|25.4|26.7|26.8|26.7|26.5|26.5|
> |8|18.8|21.2|25.5|26.6|26.8|26.6|26.5|26.5|
> |16|18.9|21.3|25.5|26.7|26.8|26.6|26.5|26.4|
> |32|19.0|21.2|25.5|26.7|26.7|26.7|26.6|26.5|
> |64|19.0|21.3|25.4|26.7|26.8|26.6|26.4|26.5|
> |128|18.9|21.3|25.4|26.7|26.9|26.7|26.5|26.4|
>
>
>
>
>
> **Table 2.b**: DTM PickScore,  Head NFE (Rows) vs. TM steps (Columns).
> |HeadNFE/TMsteps|1|2|4|8|16|32|64|128|
> |--------|----|----|----|----|----|----|----|----|
> |1|17.6|17.8|18.6|19.4|19.6|19.7|19.6|19.6|
> |2|17.7|18.3|19.7|20.6|20.9|21.0|21.0|21.0|
> |4|18.1|18.8|20.0|20.8|21.1|21.1|21.1|21.1|
> |8|18.3|18.8|20.0|20.8|21.1|21.1|21.1|21.2|
> |16|18.3|18.8|20.0|20.9|21.1|21.1|21.1|21.1|
> |32|18.3|18.8|20.0|20.9|21.1|21.1|21.1|21.1|
> |64|18.3|18.8|20.0|20.9|21.1|21.1|21.1|21.1|
> |128|18.3|18.8|20.0|20.8|21.1|21.1|21.1|21.1|
>
>
>
> **Table 3.a**: FM CLIPScore vs. Euler steps (Columns).
> |EulerSteps|1|2|4|8|16|32|64|128|
> |-----------|----|----|----|----|----|----|----|----|
> |0|15.8|16.6|19.7|23.8|25.6|25.9|25.9|26.0|
>
>
>
> **Table 3.b**: FM PickScore vs. Euler steps (Columns).
> |EulerSteps|1|2|4|8|16|32|64|128|
> |-----------|----|----|----|----|----|----|----|----|
> |0|17.9|18.0|18.7|20.0|20.8|21.0|21.0|21.0|
>
>
>
>
> **Table 4.**:DTM inference time (in seconds) for different combinations of Head NFE and TM steps on a single H100 GPU. Note that 0 head steps refers to FM. Head NFE (Rows) vs. TM steps (Columns).
> Head NFE / TM Steps | 1     | 2     | 4     | 8     | 16    | 32    | 64    | 128   |
> |------------|-------|-------|-------|-------|-------|-------|-------|--------|
> | 0          | 0.1   | 0.2   | 0.3   | 0.7   | 1.3   | 2.7   | 5.4   | 10.8   |
> | 1          | 0.1   | 0.2   | 0.4   | 0.7   | 1.4   | 2.8   | 5.6   | 11.2   |
> | 2          | 0.1   | 0.2   | 0.4   | 0.7   | 1.5   | 2.9   | 5.8   | 11.6   |
> | 4          | 0.1   | 0.2   | 0.4   | 0.8   | 1.6   | 3.1   | 6.3   | 12.5   |
> | 8          | 0.1   | 0.2   | 0.4   | 0.9   | 1.8   | 3.6   | 7.2   | 14.3   |
> | 16         | 0.1   | 0.3   | 0.6   | 1.1   | 2.2   | 4.5   | 9.0   | 17.9   |
> | 32         | 0.2   | 0.4   | 0.8   | 1.6   | 3.1   | 6.3   | 12.5  | 25.1   |
> | 64         | 0.3   | 0.6   | 1.2   | 2.5   | 4.9   | 9.9   | 19.7  | 39.4   |
> | 128        | 0.5   | 1.1   | 2.1   | 4.3   | 8.5   | 17.0  | 34.0  | 68.1   |
>
> > I’m unable to find the discussion about the limitation except line 157.
>
> More limitations appear in the conclusions section in lines (286-287). We will modify/elaborate the limitations to include “The improved performance of ARTM/FHTM comes at the price of a higher sampling cost, i.e., NFE counts are proportional to the number of transition steps, see e.g., Table 1 in the paper.”

---

> > ### Author Response · Authors · 2025-08-06
> >
> > As discussion period coming to an end - we would like to draw the reviewer's attention to the comprehensive rebuttal and new results we produced to address their comments and would greatly appreciate a response. Thanks!

---

### Note · Authors · 2025-08-12

Dear AC and reviewers,

We thank the reviewers for their insightful feedback and questions, as well as for recognizing the potential impact of our work.
We particularly appreciate the constructive engagement of reviewers jNpW and kPcj, and are pleased to have addressed all of their concerns.
We regret that reviewers ag7i and QC47 didn’t engage with our rebuttal despite the fact that we added dedicated new experiments to address their specific concerns. We hope they will have the opportunity to review them during the remaining discussion period.

We summarize the main concerns and the new experiments that address them:

1. **Time complexity of DTM:** DTM is in fact **considerably faster** than FM. Table 1 **in the response to reviewer ag7i** compares the wall-clock time of FM and DTM, and in particular DTM (16 backbone NFE and 4 head NFE) can **achieve superior performance** to FM (128 backbone NFE) with a **7x speedup**. In more detail, Tables 2(a–b) **in the response to reviewer ag7i** present the dependence of DTM’s CLIPScore and PickScore on the number of function evaluations (NFE) in the flow head and backbone. Table 4 **in the response to reviewer ag7i** reports the corresponding inference times. Tables 3(a–b) **in the response to reviewer ag7i** present flow matching’s (FM) CLIPScore and PickScore as a function of backbone NFE.

2. **More benchmarks:** In addition to the GenEval benchmark and 6 auto-metrics: CLIPScore, PickScore, ImageReward, UnifiedReward, Aesthetic, and DeQA Score, evaluated on two benchmarks: MS-COCO and PartiPrompts, we have added the benchmark suggested by Reviewer jNpW T2I-CompBench with results summarized in Table 7 **in the response to reviewer jNpW**. To summarize all the evaluations we add Table 8 **in the response to reviewer jNpW** showing the global ranking of each method (defined as the sum of its ranking across all benchmarks: GenEval, T2I-Compbench, PartiPrompts, and MS-COCO ).

3. **More baselines:** In addition to flow matching, continuous-AR, MAR and MAR-Fluid, we also added two fully discrete baselines: discrete-AR and discrete-MAR. The results are summarized in Tables 5 (a-b) and 6 **in the response to reviewer QC47**, and are consistent with the submitted paper’s claims.

We believe these additional results both strengthen our submission and address all reviewer concerns, particularly demonstrating DTM’s advantages in efficiency and performance across diverse benchmarks.

---

### Decision · Program_Chairs · 2025-09-17

**Decision:**

Accept (poster)

**Comment:**

The paper introduces Transition Matching (TM), a novel discrete-time generative modeling framework that unifies diffusion/flow models and autoregressive generation through three variants: DTM, ARTM, and FHTM. Key contributions include DTM achieving 7× speedup over flow matching while maintaining superior performance, and FHTM representing the first fully causal continuous-domain model to match non-causal methods. The comprehensive evaluation on 1.7B parameter models across multiple benchmarks with controlled experimental conditions provides strong empirical validation.

The rebuttal period was highly productive, with authors successfully addressing all major reviewer concerns through additional experiments and clarifications, leading three reviewers (QC47, jNpW, kPcj) to raise their ratings. While presentation clarity could be improved, the core technical contributions are substantial and theoretically sound.

Overall, the work advances state-of-the-art in generative modeling, opens new research directions for multimodal systems, and demonstrates significant practical improvements in both efficiency and generation quality. Hence, the AC recommends acceptance.